# Robustness in Both Domains:
# CLIP Needs a Robust Text Encoder

**Elias Abad Rocamora**[EPFL], **Christian Schlarmann**[TÜ], **Naman Deep Singh**[TÜ],

**Yongtao Wu**[EPFL], **Matthias Hein**[TÜ], **Volkan Cevher**[EPFL]

[EPFL] : LIONS - École Polytechnique Fédérale de Lausanne, Switzerland
[TÜ] : Tübingen AI center, University of Tübingen, Germany
{name.surname}@{epfl.ch, uni-tuebingen.de}

## Abstract

Adversarial input attacks can cause a significant shift of CLIP embeddings. This can affect the downstream robustness of models incorporating CLIP in the pipeline, such as text-to-image generative models or large vision language models. While some efforts have been done towards making the CLIP image encoders robust, the robustness of text encoders remains unexplored. In this work, we cover this gap in the literature. We propose LEAF: an efficient adversarial finetuning method for the text domain, with the ability to scale to large CLIP models. Our models significantly improve the zero-shot adversarial accuracy in the text domain, while maintaining the vision performance provided by robust image encoders. When combined with text-to-image diffusion models, we can improve the generation quality under adversarial noise. In multimodal retrieval tasks, LEAF improves the recall under adversarial noise over standard CLIP models. Finally, we show that robust text encoders facilitate better reconstruction of input text from its embedding via direct optimization. We open-source our code and models.

## 1 Introduction

Contrastive Language-Image Pretraining (CLIP) models embed images and captions into a shared embedding space [Radford et al., 2021]. CLIP is a simple but rather powerful tool for vision-language understanding, being employed in a wide range of multimodal tasks such as retrieval [Fang et al., 2021, Koukounas et al., 2024, Vendrow et al., 2024], Large Multimodal Models (LMMs) [Alayrac et al., 2022, Liu et al., 2023] and text-to-image generative models [Ramesh et al., 2021, Rombach et al., 2022, Ramesh et al., 2022, Podell et al., 2024].

However, the simplicity of CLIP and its plug-and-play usage becomes a double-edged sword, allowing adversarial attacks to be optimized over CLIP, and transferred to the downstream task of interest [Zhuang et al., 2023, Ghazanfari et al., 2023, 2024, Croce et al., 2025]. Recently, making the image encoder of CLIP robust has gained interest [Mao et al., 2023], making LMMs robust to adversarial perturbations by replacing the image encoder with an adversarially finetuned one [Schlarmann et al., 2024]. Nevertheless, adversarial finetuning has not been yet investigated for the text encoder.

In this work, we fill this gap by studying adversarial finetuning for CLIP text encoders, proposing *Levenshtein Efficient Adversarial Finetuning* (LEAF). Motivated by recent advancements in the image domain, we optimize the same objective as Schlarmann et al. [2024], allowing us to replace the text encoder in tasks like text-to-image generation, without needing to finetune the rest of the pipeline. Moreover, to make adversarial finetuning faster in the text domain, we propose an attack that can be parallelized within training batches, accelerating the approach of Abad Rocamora et al. [2024] by an order of magnitude with very little loss of performance.

39th Conference on Neural Information Processing Systems (NeurIPS 2025).

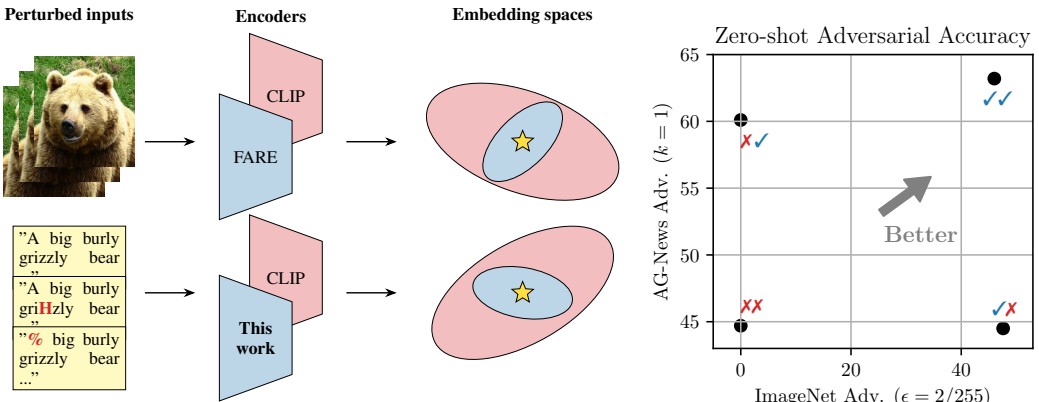

Figure 1: **Left: our idea.** Schlarmann et al. [2024] propose FARE: finetuning the CLIP image encoder to produce embeddings close to the clean image embedding (★) under image perturbations. Analogously, we finetune the CLIP *text* encoder to produce embeddings close to the clean *text* embedding (★) under *text* perturbations. **Right: results in ViT-L/14.** The first (second) ✗/✓ denotes the usage of a robust image (text) encoder. We constrain the text attacks with the Levenshtein distance and the image attacks in the $\ell_\infty$ norm. By combining the FARE robust image encoder with our robust text encoder, we obtain high adversarial accuracy in both domains.

Our models, `LEAF`, are able to improve the zero-shot adversarial accuracy of CLIP models from 44.5% to 63.3% in AG-News at distance $k = 1$ (one character change). When plugged into Stable Diffusion [Rombach et al., 2022, Podell et al., 2024], we achieve higher quality images under character-level perturbations. For retrieval tasks, our models achieve a recall 10 points higher on average than non-robust CLIP models at $k = 2$. Moreover, when inverting the embeddings of text encoders through direct optimization, we show that with `LEAF` models, we can recover a higher percentage of the original sentence. This results in `LEAF` encoders being more interpretable.

Overall, we show the robustness of CLIP text encoders can be improved with minimal effects on the clean performance in several tasks. We believe our robust CLIP models can make future models incorporating CLIP more robust and interpretable. Our code and models can be found in github.com/LIONS-EPFL/LEAF and huggingface.co/LEAF-CLIP respectively.

**Notation:** We use uppercase bold letters for matrices $\boldsymbol{X} \in \mathbb{R}^{m \times n}$, lowercase bold letters for vectors $\boldsymbol{x} \in \mathbb{R}^m$ and lowercase letters for numbers $x \in \mathbb{R}$. Accordingly, the $i^{\text{th}}$ row and the element in the $i, j$ position of a matrix $\boldsymbol{X}$ are given by $\boldsymbol{x}_i$ and $x_{ij}$ respectively. We use the operator $|\cdot|$ for the size of sets, e.g., $|\mathcal{S}(\Gamma)|$ and the length of sequences, e.g., for $\boldsymbol{X} \in \mathbb{R}^{m \times n}$, we have $|\boldsymbol{X}| = m$. For two vectors $\boldsymbol{u}, \boldsymbol{v} \in \mathbb{R}^h$, we denote the cosine similarity as $\text{sim}(\boldsymbol{u}, \boldsymbol{v}) = \frac{\boldsymbol{u}^\top \boldsymbol{v}}{||\boldsymbol{u}||_2 \cdot ||\boldsymbol{v}||_2}$. We use the shorthand $[n] = \{0, 1, \cdots, n-1\}$ for any natural number $n$.

## 2 Background

In Section 2.1 we cover the approaches improving the adversarial robustness of CLIP. In Section 2.2 we discuss robustness in the text domain.

### 2.1 Robustness of CLIP

Let $\mathcal{S}(\Gamma) = \{c_1 c_2 \cdots c_m : c_i \in \Gamma \ \forall m \in \mathbb{N} \setminus 0\}$ be the space of sequences of characters in the alphabet set $\Gamma$. We represent sentences $\boldsymbol{S} \in \mathcal{S}(\Gamma)$ as sequences of one-hot vectors, i.e., $\boldsymbol{S} \in \{0, 1\}^{m \times |\Gamma|} : ||\boldsymbol{s}_i||_1 = 1, \ \ \forall i \in [m]$. Similarly, we can represent images with $d$ pixels as real vectors $\boldsymbol{x} \in \mathbb{R}^d$. Overall, the training dataset is composed of $n$ text-image pairs $\{\boldsymbol{S}_i, \boldsymbol{x}_i\}_{i=1}^n$.

The objective of CLIP is to learn a text encoder $\boldsymbol{f_\theta} : \mathcal{S}(\Gamma) \to \mathbb{R}^h$ and an image encoder $\boldsymbol{g_\omega} : \mathbb{R}^d \to \mathbb{R}^h$, where $h$ is the embedding size and $\boldsymbol{\theta}$ and $\boldsymbol{\omega}$ are the parameters of the text and image encoders respectively. Radford et al. [2021] propose to maximize the cosine similarity of positive

sentence-image pairs relative to the cosine similarity with other sentences and images in the dataset. We denote the weights obtained after pretraining with CLIP as $\boldsymbol{\theta}_{\mathrm{CLIP}}$ and $\boldsymbol{\omega}_{\mathrm{CLIP}}$.

In order to make the image encoder $\boldsymbol{g}_{\boldsymbol{\omega}}$ robust in the zero-shot classification task, Mao et al. [2023] use the sentences $\boldsymbol{S}_j =$ "a photo of a $\mathrm{LABEL}_j$," $\forall j \in [o]$, where $o$ is the number of classes. Then, given a dataset of images and labels $\{\boldsymbol{x}_i, y_i\}_{i=1}^n$, so that $y_i \in [o]$, Mao et al. [2023] optimize:

$$\min_{\boldsymbol{\omega}} \sum_{i=1}^n \max_{||\boldsymbol{\delta}_i||_\infty \leq \epsilon} -\log\left(\frac{e^{\boldsymbol{f}_{\boldsymbol{\theta}_{\mathrm{CLIP}}}(\boldsymbol{S}_{y_i})^\top \boldsymbol{g}_{\boldsymbol{\omega}}(\boldsymbol{x}_i + \boldsymbol{\delta}_i)}}{\sum_{j=1}^o e^{\boldsymbol{f}_{\boldsymbol{\theta}_{\mathrm{CLIP}}}(\boldsymbol{S}_j)^\top \boldsymbol{g}_{\boldsymbol{\omega}}(\boldsymbol{x}_i + \boldsymbol{\delta}_i)}}\right). \qquad \text{(TeCoA)}$$

TeCoA significantly improves the robustness of the image encoder. However, it generalizes poorly to image classification tasks that are not part of the fine-tuning dataset, and degrades the performance when employed in an LMM pipeline, as shown by Schlarmann et al. [2024]. In order to overcome this, Schlarmann et al. [2024] propose FARE, which intends to preserve the original image embeddings while being robust. To do so, they optimize:

$$\min_{\boldsymbol{\omega}} \sum_{i=1}^n \max_{||\boldsymbol{\delta}_i||_\infty \leq \epsilon} ||\boldsymbol{g}_{\boldsymbol{\omega}_{\mathrm{CLIP}}}(\boldsymbol{x}_i) - \boldsymbol{g}_{\boldsymbol{\omega}}(\boldsymbol{x}_i + \boldsymbol{\delta}_i)||_2^2. \qquad \text{(FARE)}$$

The FARE objective allows to employ the obtained image encoder within an LMM pipeline with minimal clean performance degradation. Motivated by these findings, in this work we construct a similar loss in the text domain (Eq. (`TextFARE`)) and adapt the algorithm to the challenges of this new domain (`LEAF`). See Fig. 1 for a visualization of the FARE and `LEAF` approaches.

## 2.2 Robustness in the text domain

Belinkov and Bisk [2018], Alzantot et al. [2018] showed that text classifiers are not robust to natural or adversarial noise, with text adversarial attacks being used in Large Language Models [Zou et al., 2023] and text-to-image generative models [Zhang et al., 2025]. Generally, given a sentence $\boldsymbol{S}$, a model $\boldsymbol{f}$ and some loss function $\mathcal{L}$, the adversarial attack problem can be formulated as:

$$\max_{\boldsymbol{S}' \in \mathcal{N}(\boldsymbol{S})} \mathcal{L}(\boldsymbol{f}(\boldsymbol{S})),$$

where $\mathcal{N}(\boldsymbol{S})$ is a set of neighboring sentences, i.e., the threat model. A great challenge in the text domain is defining a valid threat model, as the semantics of the sentence $\boldsymbol{S}$ should be preserved according to the task [Morris et al., 2020]. In the literature, we can categorize adversarial attacks into two main threat models: *token* and *character* level attacks. With token level attacks set to replace/insert/delete a small number of tokens in the sentence [Ren et al., 2019, Jin et al., 2020, Li et al., 2019, Garg and Ramakrishnan, 2020, Lee et al., 2022, Ebrahimi et al., 2018, Li et al., 2020, Guo et al., 2021, Hou et al., 2023]. Similarly, character-level attacks replace/insert/delete a small number of characters in the sentence [Belinkov and Bisk, 2018, Ebrahimi et al., 2018, Gao et al., 2018, Pruthi et al., 2019, Yang et al., 2020, Liu et al., 2022, Abad Rocamora et al., 2024]. Both approaches can be thought of as keeping a small Levenshtein distance [Levenshtein, 1966] between the original and attacked sentences in the token or character-level.

**Semantic constraints:** To ensure that semantics are preserved, token-level attacks usually constrain $\mathcal{N}(\boldsymbol{S})$ further by only allowing token replacements between tokens with high similarity in the embedding space [Jin et al., 2020]. But, even with such semantic constraints, several works have pointed out that token level attacks do not preserve semantics [Morris et al., 2020, Dyrmishi et al., 2023], with Hou et al. [2023] reporting $56.5\%$ of their attacks change the semantics of the sentence. Due to the difficulty in preserving semantics, we focus on character-level attacks in this work.

In the case of the character-level attacks, to further preserve semantics and simulate natural typos, some works constrain the attack to only replace characters that are nearby in the English keyboard [Belinkov and Bisk, 2018, Huang et al., 2019]. Others do not allow the attack to modify the first and last letter of words, to perturb short words, to perturb the same word twice or to insert special characters [Pruthi et al., 2019, Jones et al., 2020]. In the context of text-to-image generation, Chanakya et al. [2024] find that changing one character in the sentence can change one word for another and the text-to-image model accordingly generates a different object in the image. To avoid this, Chanakya et al. [2024] introduce the semantic constraint of not allowing new English words to appear after the attack. In this work, we decide to adopt the semantic constraints of [Chanakya et al., 2024] and find they are especially useful when performing adversarial finetuning of the CLIP text encoders, see Section 4.2.2.

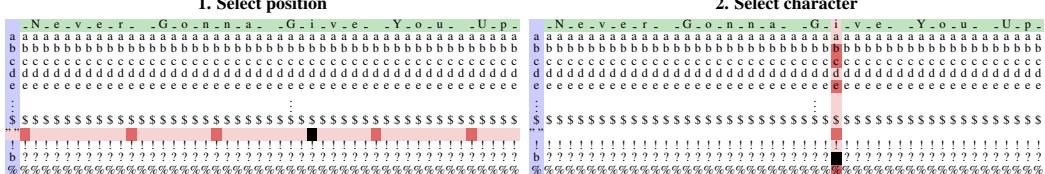

**1. Select position**   **2. Select character**

**3. Final perturbation:** "Never Gonna G?ve You Up"

Figure 2: **Schematic and example of the attack used in** `LEAF`**:** In the first step, we randomly select $\rho = 6$ positions, replace these with a whitespace and select the position with the highest loss. Next, we randomly select $\rho$ characters from $\Gamma$, replace them in the chosen position and choose the one with the highest loss as the final perturbation. During training, the attack evaluates $\rho \times B$ sentences in every forward pass, where $B$ is the batch size. For more details, see Algorithm 1 in the appendix.

## 3 Method

In order to make the text encoder adversarially robust, we extend Eq. (FARE) to the text domain as:

$$\min_{\boldsymbol{\theta}} \sum_{i=1}^{n} \max_{\boldsymbol{S}'_i : d_{\mathrm{Lev}}(\boldsymbol{S}_i, \boldsymbol{S}'_i) \leq k \wedge \boldsymbol{S}'_i \in \mathcal{C}(\boldsymbol{S}_i)} ||\boldsymbol{f}_{\boldsymbol{\theta}_{\mathrm{CLIP}}}(\boldsymbol{S}_i) - \boldsymbol{f}_{\boldsymbol{\theta}}(\boldsymbol{S}'_i)||_2^2 , \qquad \text{(TextFARE)}$$

where the Levenshtein $d_{\mathrm{Lev}}$ distance is bounded by a parameter $k$, and $\mathcal{C}(\boldsymbol{S})$ is either the complete set of sentences $\mathcal{S}(\Gamma)$ or a subset only containing sentences with semantic constraints, see Section 2.2.

Intuitively, if the original CLIP encoder evaluated at the original sentence ($\boldsymbol{f}_{\boldsymbol{\theta}_{\mathrm{CLIP}}}(\boldsymbol{S})$) provides a good performance in downstream tasks, e.g., zero-shot classification or text-to-image generation, then, by solving Eq. (TextFARE), we will obtain a model that achieves similar performance under perturbations of the sentence. Moreover, Eqs. (FARE) and (TextFARE) allow for decoupled training of the text and image encoders.

Motivated by Danskin's Theorem [Danskin, 1966, Latorre et al., 2023], we can (approximately) solve min-max problems by maximizing the inner problem and minimizing the error on the obtained perturbation. In the case of Eq. (FARE), Projected Gradient Descent (PGD) is used for the inner maximization problem [Madry et al., 2018, Schlarmann et al., 2024]. Similarly, we can use any adversarial attack to maximize the inner problem in Eq. (TextFARE), e.g., Gao et al. [2018], Abad Rocamora et al. [2024].

However, not every attack is adequate for adversarial finetuning, e.g., in the image domain, the strongest attacks in the AutoAttack ensemble [Croce and Hein, 2020] are never used during training due to their expensive time requirements. Contrarily, cheaper PGD attacks are used during training, providing fast training and generalization to stronger adversarial attacks Goodfellow et al. [2015], Madry et al. [2018], Shafahi et al. [2019], Wong et al. [2020]. The desiderata for an adversarial attack used during training can be captured by two points: *(i) High adversarial robustness to strong attacks after training, (ii) Low computational resources*.

As a baseline attack in the text domain, we select Charmer [Abad Rocamora et al., 2024]. Adversarial training with Charmer in text classification results in strong adversarial robustness, satisfying (i). Nevertheless, Charmer is not resource-efficient during training and thereby does not satisfy our second desiderata (ii). This is due to Charmer needing to evaluate a number of perturbations $\mathcal{O}((2 \cdot |\boldsymbol{S}_i| + 1) + n_{\mathrm{Charmer}} \cdot |\Gamma|)$, which depends on the length of the sentence being attacked. This makes it harder to perform the attack simultaneously over sentences in a batch.

Overcoming this limitation, we propose *Levenshtein Efficient Adversarial Finetuning* (`LEAF`): utilizing a training-time attack that evaluates a constant number of perturbations $\rho$ per sentence. Our attack replaces a test character (the whitespace) in $\rho$ random positions within the sentence to select the position with the highest loss. Then, $\rho$ random characters are replaced in the chosen position to choose again the one with the highest loss. Overall, this allows to perform the attack in two sequential evaluations of $B \cdot \rho$ sentences, where $B$ is the batch size. A visual representation of our attack is available in Fig. 2. Interestingly, if $\rho = 1$, our attack performs a random perturbation. For a more detailed discussion on `LEAF`, we refer to Appendix B. In Section 4.2 we empirically show `LEAF` satisfies our two desiderata.

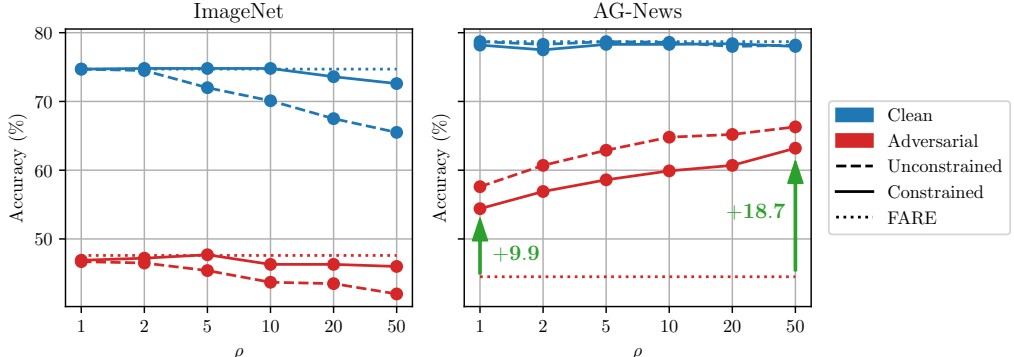

Figure 3: **Training hyperparameter effects:** We report the zero-shot clean and adversarial accuracy in the image (ImageNet) and text (AG-News) domains with FARE as a baseline. When no semantic constraints are employed (Section 2.2), the robustness in the text domain is improved at the cost of significantly degrading the image domain performance. Adding semantic constraints improves the robustness in the text domain with minimal effects on the image domain. Using random perturbations ($\rho = 1$) improves the AG-News adversarial accuracy by 9.9 points, with stronger attacks ($\rho = 50$) providing the best performance with 18.7 points of improvement.

## 4 Experiments

We start by introducing our experimental setup in Section 4.1. In Section 4.2 we cover our training results and display the interplay between $\rho$, $k$ and the usage of additional constraints during training. In Section 4.3 we present the performance of our models in zero-shot classification. In Section 4.4, we evaluate our CLIP models in multimodal retrieval tasks. In Section 4.5 we evaluate the performance of our CLIP text encoders when incorporated into text-to-image generative models. Finally, in Section 4.6 we evaluate how amenable our models are to embedding inversion. Additional experiments, including an evaluation with token-level attacks, are available in Appendix D.

### 4.1 Experimental setup

We train our text encoders for 30 epochs on the first $80,000$ samples of the DataComp-small dataset [Gadre et al., 2023] with a batch size of 128 sentences, $k = 1, \rho = 50$ and semantic constraints, see Section 4.2.2, employing CLIP-ViT-L/14, OpenCLIP-ViT-H/14, OpenCLIP-ViT-g/14 and OpenCLIP-ViT-bigG/14 models. On the visual side, we scale the training method of Schlarmann et al. [2024] to ViT-H/14 and ViT-g/14, using an $\ell_\infty$ threat model with radius $\epsilon = 2/255$. See Appendix B.3 for a detailed account of hyperparameters. For evaluating the adversarial robustness with respect to image perturbations, we follow Schlarmann et al. [2024] and employ the first two APGD attacks from the AutoAttack ensemble [Croce and Hein, 2020] with $\epsilon = 2/255$. In the text domain, we choose Charmer-20 with $k = 1$ [Abad Rocamora et al., 2024] for evaluation. We employ the semantic constraints considered by [Chanakya et al., 2024] in the text-to-image and retrieval tasks. For the zero shot classification tasks, we do not employ such constraints as done by Abad Rocamora et al. [2024]. For a discussion on the use of constraints, we refer to Appendix D.1. For zero shot sentence classification with CLIP models, we follow the setup of Qin et al. [2023], see Appendix B.4 for more details. For additional details, we refer to Appendix D.

### 4.2 Training robust text encoders

In Section 4.2.1 we analyze the performance and training speed of Charmer and `LEAF`. In Section 4.2.2 we analyze how the performance is affected by our hyperparameters, i.e., $k$, $\rho$ and $\mathcal{C}(\boldsymbol{S})$.

### 4.2.1 Faster adversarial finetuning

First, we evaluate the performance of `LEAF` in terms of time and adversarial accuracy against training with Charmer [Abad Rocamora et al., 2024] with $n_{\text{Charmer}} \in \{1, 20\}$. To do so, we train CLIP-ViT-

Table 1: **Selecting the best attack for Adversarial Finetuning on ViT-B-32:** We measure the AG-News clean (Acc.) and adversarial accuracy (Adv.) at $k = 1$ with Charmer-20 and the time in seconds to attack a batch of 128 sentences. We perform Adversarial Finetuning (Eq. (`TextFARE`)) for 1 epoch with $k = 1$ using the attacks Charmer-1, Charmer-20 and `LEAF` with $\rho \in \{20, 50\}$. Our approach minimally affects the adversarial accuracy while being an order of magnitude faster than the fastest Charmer variant.

| Defense | AG-News | | Time (s) |
| --- | --- | --- | --- |
| | Acc. (%) | Adv. (%) | |
| Charmer-20 | $76.70_{(\pm 0.14)}$ | $60.17_{(\pm 0.31)}$ | $118.19_{(\pm 53.68)}$ |
| Charmer-1 | $76.37_{(\pm 0.21)}$ | $\mathbf{60.20}_{(\pm 0.37)}$ | $15.17_{(\pm 28.98)}$ |
| LEAF ($\rho = 50$) | $76.63_{(\pm 0.21)}$ | $59.80_{(\pm 0.37)}$ | $3.23_{(\pm 0.17)}$ |
| LEAF ($\rho = 20$) | $\mathbf{76.87}_{(\pm 0.25)}$ | $58.30_{(\pm 0.29)}$ | $\mathbf{1.83}_{(\pm 0.11)}$ |

B-32 for 1 epoch at $k = 1$ and using $\rho \in \{20, 50\}$ for `LEAF` over three random training seeds. We measure the clean and adversarial accuracies with Charmer-20 on AG-News [Gulli, 2005, Zhang et al., 2015] and the average time to attack a batch of 128 samples.

In Table 1 we can observe that `LEAF` attains comparable clean and adversarial accuracies in comparison to the Charmer variants, while being significantly faster, i.e., $1.83$ and $3.23$ seconds per batch for our method in comparison to $15.17$ and $118.19$ seconds for the Charmer variants.

### 4.2.2 The effect of our hyperparameters

In order to test the influence of our training hyperparameters, we finetune CLIP-ViT-L/14 initialized from pretrained FARE weights [Schlarmann et al., 2024] with $\rho \in \{1, 2, 5, 10, 20, 50\}$, $k \in \{1, 2\}$ and $\mathcal{C}(\boldsymbol{S})$ including and not including semantic constraints. To evaluate how our method improves the robustness in the text domain, and affects the robustness in the image domain, we measure the clean and adversarial accuracies on ImageNet and AG-News.

In Fig. 3 we report the performance for $k = 1$. When increasing $\rho$, the adversarial accuracy in the text domain increases consistently. However, when employing unconstrained training attacks, both the clean and adversarial performance in the image domain are significantly degraded, e.g. at $\rho = 50$, a clean accuracy of $65.5\%$ vs. $74.7\%$ for the FARE model. In contrast, when applying semantic constraints, the improvements in robustness in the text domain follow a similar trend and the performance in the image domain is less degraded. For $k = 2$, we can extract the same insights, see Fig. 8. Overall, we select $\rho = 50, k = 1$ and the use of semantic constraints during training.

### 4.3 Zero-shot classification

We show the ImageNet and AG-News performance of the models when using robust encoders in image and/or text domain in Table 2 and Fig. 1. We observe that our robust text encoders introduce only minimal drop in image performance, while significantly improving the robustness in the text domain. Moreover, we observe that the effectiveness of FARE for fine-tuning robust image encoders that was demonstrated for ViT-L/14 by Schlarmann et al. [2024], extends to the larger ViT-H/14 and ViT-g/14 models. The lower performance of ViT-g/14 on ImageNet could be attributed to the smaller training batch size, see Appendix B.3. Importantly, only models that use a robust encoder in both domains achieve robustness in both tasks.

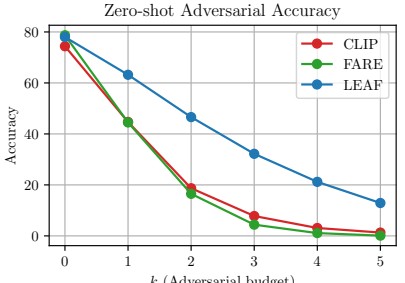

Figure 4: **Larger perturbations:** We evaluate the adversarial accuracy in AG-News for $k \in \{1, 2, 3, 4, 5\}$ in the ViT-L/14 scale. Our model (`LEAF`) obtains the highest adversarial accuracy at all values of the distance bound $k$.

In Fig. 4 we report the adversarial accuracy of the ViT-L/14 sized models in the AG-News dataset for $k \in \{0, 1, 2, 3, 4, 5\}$, with $k = 0$ representing the clean accuracy. Our model, while being trained with $k = 1$, is able to extrapolate the robustness to larger $k$. We observe that the CLIP and FARE models obtain a nearly zero adversarial accuracy for $k \geq 4$, while our model, is able to obtain the highest performance for any $k$.

Table 2: **Zero-shot classification.** We report the adversarial accuracy (Adv.) on ImageNet with the first two attacks of AutoAttack (APGD-CE, APGD-t) at $\epsilon = 2/255$ and on AG-News with Charmer-20 at $k = 1$. Only models employing robust image *and* text encoders are robust in both domains.

| Robust Encoder | | CLIP-ViT-L/14 | | | | OpenCLIP-ViT-H/14 | | | | OpenCLIP-ViT-g/14 | | | |
| | | ImageNet | | AG-News | | ImageNet | | AG-News | | ImageNet | | AG-News | |
| Image | Text | Acc. | Adv. | Acc. | Adv. | Acc. | Adv. | Acc. | Adv. | Acc. | Adv. | Acc. | Adv. |
| ✗ | ✗ | 76.4 | 0.0 | 74.4 | 44.7 | 77.2 | 0.0 | 71.1 | 37.6 | 77.8 | 0.0 | 67.3 | 35.8 |
| ✓ | ✗ | 74.7 | 47.6 | 78.7 | 44.5 | 76.8 | 48.4 | 70.7 | 37.5 | 73.8 | 41.8 | 66.4 | 32.9 |
| ✗ | ✓ | 73.4 | 0.0 | 73.9 | 60.1 | 77.0 | 0.0 | 71.1 | 50.2 | 76.3 | 0.0 | 67.3 | 47.4 |
| ✓ | ✓ | 72.6 | 46.0 | 78.0 | 63.2 | 76.8 | 46.3 | 72.3 | 53.3 | 72.0 | 41.3 | 66.7 | 46.3 |

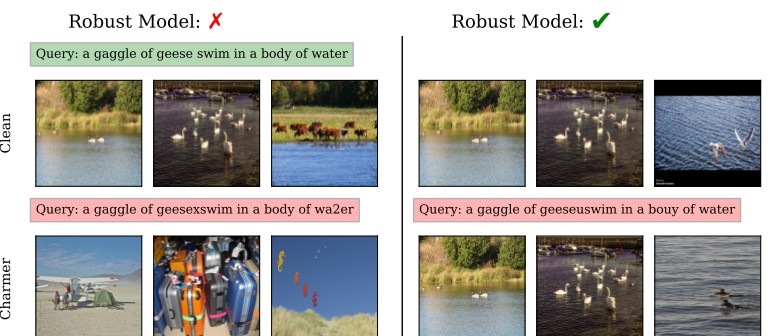

Figure 5: **Visualizing MS-COCO retrieved images.** For our ViT-L/14 robust model and its non-robust counterpart, we show the top-3 retrieved images for the original Query and the perturbed Query via Charmer ($k = 2, n = 10$) attack. The robust model is able to preserve the order and retrieves semantically relevant images even for the perturbed query. More illustrations can be found in Appendix D.5. The target query in this case was "This is an image of a pyramid".

## 4.4 Text-image retrieval

Robustness of CLIP models to perturbations of textual queries is important as these models are often used as dataset/content filters Hong et al. [2024] and NSFW detectors Schuhmann et al. [2022], meaning any false negative can be detrimental. The robustness of retrieval based filters for visual adversaries has already been tested in Croce et al. [2025]. Consider the case where a CLIP based NSFW filter is queried with a perturbed query, any false negative retrieval here would detrimental and concerning. To test how robust CLIP models are to such character based queries in retrieval setup, we test on the MS-COCO dataset as a proxy task.

For $1,000$ validation set queries, the attack maximizes the similarity between the test query and a target string using different variants of the Charmer attack. Given some query text $\boldsymbol{S}$ and corresponding embedding $\boldsymbol{f_\theta}(\boldsymbol{S})$, we maximize the cosine similarity between $\boldsymbol{f_\theta}(\boldsymbol{S})$ and $\boldsymbol{f_\theta}(\boldsymbol{T})$, where $\boldsymbol{T}$ is a target text semantically unrelated to $\boldsymbol{S}$. The objective takes the following form,

$$\max_{\boldsymbol{S}':d_{\mathrm{Lev}}(\boldsymbol{S},\boldsymbol{S}')\leq k \wedge \boldsymbol{S}'\in\mathcal{C}(\boldsymbol{S})} \mathrm{sim}\left(\boldsymbol{f_\theta}(\boldsymbol{S}'), \boldsymbol{f_\theta}(\boldsymbol{T})\right). \tag{1}$$

The optimization is done with the constrained Charmer attack for a different number of character changes. $\boldsymbol{S}'$ is initialized with $\boldsymbol{S}$, and the overall perturbation set is constrained with $\mathcal{C}(\boldsymbol{S})$ from Chanakya et al. [2024]. The formulation of the attack above can be seen as a targeted attack, the same attack can be done in an untargeted manner as in Eq. (2).

In Table 3, for different CLIP models, we show average *Recall* across 3 target strings, detailed results for each target can be found in Appendix D.5. For both 1 ($k = 1$) and 2 ($k = 2$) character perturbations, we see that the non-robust CLIP models retrieval performance goes down. Our robust models on the other hand showcase strong robustness while showing a small degradation in clean performance. For LEAF, the clean performance follows a trade-off with robustness depending on $\rho$, see Appendix D.5. Fig. 5, visualizes the attack and the top-3 retrieved images for a sample test query. Under perturbation, the non-robust model retrieves completely irrelevant images. The robust

Table 3: **MS-COCO text-to-image retrieval:** The statistics of the targeted Charmer adversarial attack (with $k = 1, 2$ and semantic constraints) are averaged over 3 target strings. ✗: denotes a non-robust CLIP model, whereas ✓ indicates CLIP model robust in both image and text domains.

| Model | Robust | Clean Recall@1 | Clean Recall@5 | Eval. $k$ | Charmer-Con Recall@1 | Charmer-Con Recall@5 |
|---|---|---|---|---|---|---|
| CLIP-ViT-L/14 | ✗ | 49.11 | 73.79 | 1 | 37.31 | 62.67 |
| | | | | 2 | 30.66 | 52.76 |
| | ✓ | 48.71 | 73.71 | 1 | 45.06 | 69.35 |
| | | | | 2 | 40.22 | 65.09 |
| OpenCLIP-ViT-H/14 | ✗ | 58.64 | 81.29 | 1 | 47.81 | 72.22 |
| | | | | 2 | 39.26 | 63.35 |
| | ✓ | 56.80 | 80.65 | 1 | 52.97 | 77.26 |
| | | | | 2 | 49.31 | 73.50 |
| OpenCLIP-ViT-g/14 | ✗ | 60.64 | 82.22 | 1 | 47.93 | 72.71 |
| | | | | 2 | 37.51 | 61.82 |
| | ✓ | 55.98 | 79.33 | 1 | 52.30 | 76.95 |
| | | | | 2 | 48.71 | 73.71 |

model on the other hand, preserves the order and retrieves images relevant to the query. Moreover, in almost all cases it retrieves the top-1 image correctly, see Appendix D.5 for more such examples. Starting with $k = 1$ text perturbations, we test the robustness of different variants of CLIP-ViT-L/14 models to bimodal attacks using APGD for image perturbations. Even in this more challenging setup, LEAF attains the most robust models, without sacrificing clean performance. We defer the associated results and discussion to Appendix D.5.1.

### 4.5 Robustness of text-to-image models

In this section, we evaluate the performance of our robust text encoders when plugged into text-to-image generation pipelines. We take SD-1.5 [Rombach et al., 2022] and SDXL [Podell et al., 2024]. SD-1.5 employs the text encoder from ViT-L/14 and SDXL employs two text encoders: from ViT-L/14 and ViT-bigG/14. In order to attack the model, we follow Zhuang et al. [2023] by only accessing the text encoder. Given a sentence $S$, we employ Charmer-20 to solve:

$$\min_{S':d_{\mathrm{Lev}}(S,S')\leq k \wedge S'\in\mathcal{C}(S)} \mathrm{sim}(f_{\theta}(S), f_{\theta}(S')). \tag{2}$$

By minimizing the similarity between the original and perturbed embedding, we expect that the model generates images that do not align to the original caption. For SDXL, we maximize the average dissimilarities for both encoders. To analyze the quality of the generated images, through CLIP-ViT-B-16, we measure the CLIPScore between the original caption $S$ and the generated image. In Fig. 6 we present the MS-COCO [Lin et al., 2014] SDXL image generation results. We can observe that the CLIPScore of SDXL with the LEAF encoders is significantly larger than the original SDXL for $k \geq 1$. On the right-hand-side of Fig. 6 we present the generated images for the first five captions in the MS-COCO validation dataset at $k = 2$, where for two captions, the original SDXL model produces completely different images compared to the original ones.

In Appendix D.3 we include additional text-to-image generation details and experiments over SD-1.5 and FLUX.1-dev [Black Forest Labs et al., 2025]. Interestingly, the generation quality of FLUX.1-dev can be severely degraded when only attacking its CLIP ViT-L/14 text encoder, see Table 13. We observe that the most common attack when the word "woman" appears, consists of replacing the final "n" for another character, see Table 19. This leads FLUX.1-dev to produce images of snakes as the tokens of the word "woma", a python species (Woma python), appear in the sentence. In Fig. 7 we report the images generated with FLUX.1-dev with the original CLIP encoder and the LEAF counterpart over 10 random seeds. When using our text encoder, the model is able to distinguish based on the rest of the sentence, whether a "woman" or a "woma" should be generated.

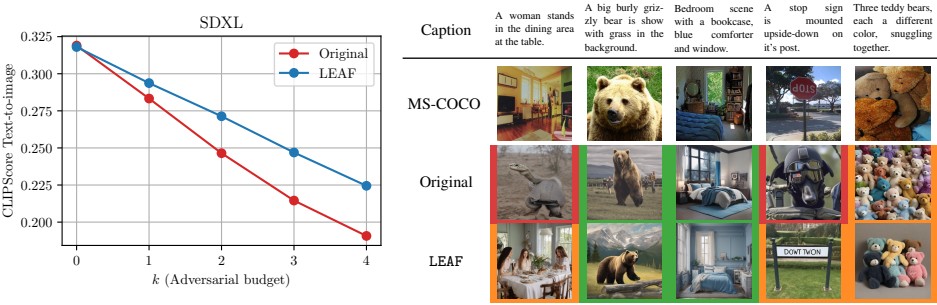

Figure 6: **Text-to-image generation results on SDXL:** On the left side, we present the MS-COCO CLIPScores of SDXL. The LEAF text encoders consistently improve the generation quality of SDXL under adversarial noise. On the right, we present the first five MS-COCO samples from the validation set and the corresponding SDXL generations at $k = 2$. The color borders indicate null, partial and total matching to the original image. With the original encoder, images 1 and 4 do not match at all the original ones. With the FARE encoders, all of the five images resemble the original ones, with some errors like the mismatch in the number of objects in image 5.

Table 4: **Text embedding inversion.** We invert text embeddings and measure the quality of reconstructions with various metrics. Robust models yield better reconstructions according to all metrics.

| Model | Robust | sim $\uparrow$ | Word Rec. $\uparrow$ | Token Rec. $\uparrow$ | BLEU $\uparrow$ |
|---|---|---|---|---|---|
| CLIP-ViT-L/14 | ✗ | 0.89 | 34.4 | 38.9 | 8.3 |
|  | ✓ | 0.95 | 46.4 | 52.0 | 12.2 |
| OpenCLIP-ViT-H/14 | ✗ | 0.86 | 33.5 | 34.1 | 8.9 |
|  | ✓ | 0.93 | 49.0 | 50.3 | 13.7 |
| OpenCLIP-ViT-g/14 | ✗ | 0.94 | 43.7 | 48.1 | 5.6 |
|  | ✓ | 0.96 | 54.8 | 60.6 | 12.2 |

## 4.6 Text embedding inversion

It is well known that robust models in the vision domain possess more interpretable gradients than clean models [Santurkar et al., 2019], which can be exploited to generate visual counterfactual explanations [Augustin et al., 2020, Boreiko et al., 2022]. Moreover, this allows to reconstruct images from their embeddings of a robust model by direct gradient based optimization [Croce et al., 2025].

We test if this advantageous property of robust vision models also holds in robust text models. To this end, we study the ability to invert text embeddings. Given an embedding $f_\theta(S)$, the goal is to reconstruct the unknown text $S$. Therefore we aim to solve the objective

$$\max_{S' \in \mathcal{S}(\Gamma)} \text{sim}\left(f_\theta(S'), f_\theta(S)\right). \tag{3}$$

To this end, we use the optimization method from Wen et al. [2023], where the text is initialized uniformly at random over the vocabulary of tokens and optimized via a gradient based algorithm.

We randomly sample 100 captions from MS-COCO, embed them via the given original and robust text encoders, and measure the success of reconstruction with four metrics: The cosine similarity between $f_\theta(S')$ and $f_\theta(S)$, i.e., the objective in Eq. (3). *Word Recall* and *Token Recall* are the percentages of words/tokens in the original text that appear in the reconstruction, irrespective of order. Finally, BLEU [Papineni et al., 2002] is an ordering-aware similarity metric.

We show results in Table 4. The models with robust text encoders are best in every metric. Interestingly, we observe that the reconstructions of robust models generally improve when scaling up model size, while for non-robust models it does not improve from ViT-L/14 to ViT-H/14, but improves from ViT-H/14 to ViT-g/14. We observe that BLEU scores are low for all models, indicating that while many words are reconstructed correctly, their ordering is not. This could be attributed to the bag-of-words behavior of CLIP models discovered by Yüksekgönül et al. [2023]. We show some randomly selected example reconstructions in Appendix Tables 22 and 23.

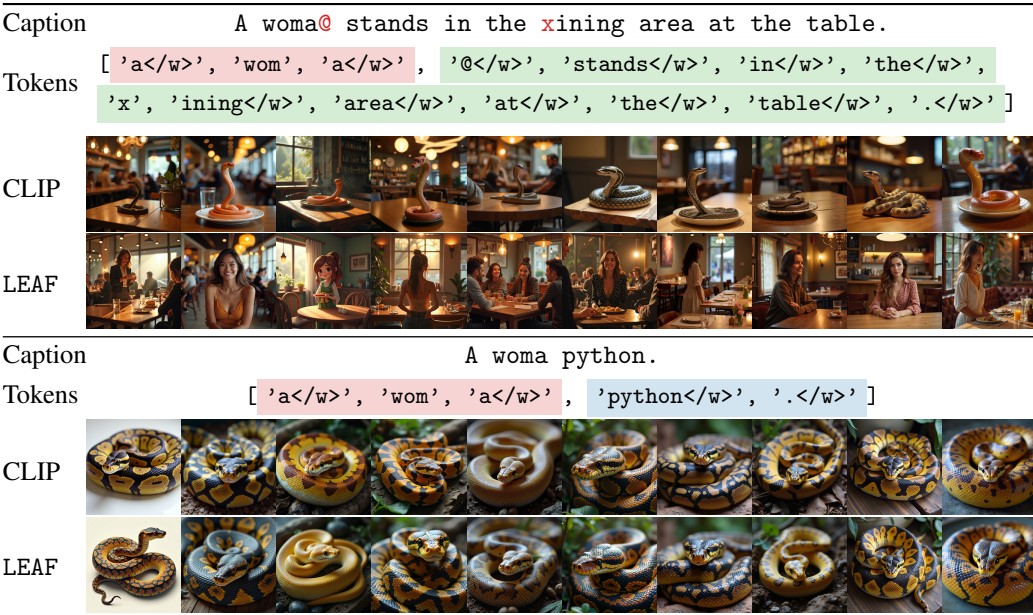

| Caption | A woma@ stands in the xining area at the table. |
|---------|--------------------------------------------------|
| Tokens | [ 'a</w>', 'wom', 'a</w>' , '@</w>', 'stands</w>', 'in</w>', 'the</w>', 'x', 'ining</w>', 'area</w>', 'at</w>', 'the</w>', 'table</w>', '.</w>' ] |
| CLIP | |
| LEAF | |

| Caption | A woma python. |
|---------|----------------|
| Tokens | [ 'a</w>', 'wom', 'a</w>' , 'python</w>', '.</w>' ] |
| CLIP | |
| LEAF | |

Figure 7: **Text-to-image generation with FLUX.1-dev:** We generate images with 10 random seeds using the original CLIP ViT-L/14 text encoder and the LEAF variant. The model using the CLIP text encoder consistently generates snakes for the first sentence, probably due to the appearance of the word "woma", a kind of snake (Woma python). When using our robust text encoder, we can accurately generate a woman and are also able to generate woma pythons when prompted to do so. While both captions start with ▮, our text encoder distinguishes between the ▮ and ▮ continuations.

## 5 Conclusion

This work takes a first, systematic step toward *bimodal* robustness of CLIP by addressing the long-neglected text side. We introduced LEAF, a simple and efficient adversarial fine-tuning scheme for text encoders that mirrors the FARE philosophy on the image side: preserve the location of the clean embedding while enforcing invariance to small perturbations. For our adversarial fine-tuning scheme we develop a training-time character-level attack that allows for efficient training. In doing so, we showed that robustness in the text domain is both practically achievable and practically useful. Across zero-shot classification, text-to-image retrieval, and text-to-image generation, LEAF improves robustness to character-level attacks consistently, while leaving the clean performance intact.

Importantly, we show that robust CLIP text encoders obtained via LEAF can be combined with robust CLIP image encoders (e.g. FARE) to yield CLIP models that are robust on both input domains. This yields the first recipe that *jointly* elevates robustness in both modalities, and it scales without bespoke architectural changes or heavy joint training. Moreover, the method is modular: encoders can be swapped without touching downstream models, e.g. in text-to-image pipelines.

Notably, while we focus the empirical evaluation in this work on CLIP based models, our LEAF method could be applied to any text encoder: see Table 27 for an illustrative example beyond CLIP, where a BERT model is finetuned for sentence classification.

**Limitations:** Our robust image and text encoders are finetuned in isolation, joint training could yield larger robustness gains at higher training cost. Nevertheless, our bimodally robust models are validated against inference-time attacks that optimize over both modalities (see Table 25). In this work, we did not train models to be robust to token-level attacks, as these attacks often change the semantics of sentences [Dyrmishi et al., 2023]. Due to computational constraints, we did not train the largest image encoders (OpenCLIP-ViT-bigG) or the largest EVA-CLIP models [Sun et al., 2024]. Our approach has not yet been tested in other tasks using text encoders, e.g., RAG [Lewis et al., 2020]. We hope that our paper fosters advances in these areas.

## Acknowledgments

We thank the NeurIPS 2025 organization committee and reviewers for their work. This work was supported by the Swiss National Science Foundation (SNSF) under grant number 200021_205011. Research was sponsored by the Army Research Office and was accomplished under Grant Number W911NF-24-1-0048. This work was supported by Hasler Foundation Program: Hasler Responsible AI (project number 21043). This work was supported as part of the Swiss AI Initiative by a grant from the Swiss National Supercomputing Centre (CSCS) under project ID a07 on Alps. EAR, YW and VC thank Gosia Baltaian for her administrative help. We thank the International Max Planck Research School for Intelligent Systems (IMPRS-IS) for supporting CS and NDS. We acknowledge support from the Deutsche Forschungsgemeinschaft (DFG, German Research Foundation) under Germany's Excellence Strategy (EXC number 2064/1, project number 390727645), as well as in the priority program SPP 2298, project number 464101476. Any opinions, findings, and conclusions or recommendations expressed in this material are those of the authors and do not necessarily reflect the views of the sponsors.

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

## A  Broader impact

This work positively impacts society by strengthening models that employ CLIP text encoders against perturbations in the text input, which is particularly important for safety-critical and high-volume applications. Practitioners can harden existing CLIP-based systems by adopting our adversarially robust text encoders as drop-in replacements with minimal changes. We provide source code and open source models to support responsible deployment.

## B  Additional details

In this section, we provide additional details on the implementation of our method and the experimental setting.

**Additional Notation:** Given two matrices $A \in \mathbb{R}^{m \times d}$ and $B \in \mathbb{R}^{n \times d}$, we define $A \oplus B = \begin{bmatrix} A \\ B \end{bmatrix} \in \mathbb{R}^{(m+n) \times d}$. Concatenating with the empty sequence $\emptyset$ results in the identity $A \oplus \emptyset = A$. We denote as $A_{2:} \in \mathbb{R}^{(m-1) \times d}$ the matrix obtained by removing the first row.

### B.1  Method details

Firstly, we characterize the single-character perturbations following Abad Rocamora et al. [2024].

**Definition B.1** (Expansion and contraction operators). Let $\mathcal{S}(\Gamma)$ be the space of sentences with alphabet $\Gamma$ and the special character $\xi \notin \Gamma$, the pair of expansion-contraction functions $\phi : \mathcal{S}(\Gamma) \to \mathcal{S}(\Gamma \cup \{\xi\})$ and $\psi : \mathcal{S}(\Gamma \cup \{\xi\}) \to \mathcal{S}(\Gamma)$ is defined as:

$$\phi(S) := \begin{cases} \xi & \text{if } |S| = 0 \\ \xi, S_1 \oplus \phi(S_{2:}) & \text{otherwise} . \end{cases} \quad \psi(S) := \begin{cases} \emptyset & \text{if } |S| = 0 \\ \psi(S_{2:}) & \text{if } S_1 = \xi \\ S_1 \oplus \psi(S_{2:}) & \text{otherwise} . \end{cases}$$

Clearly, $\phi(S)$ aims to insert $\xi$ into $S$ in all possible positions between characters and at the beginning and end of the sentence, and thus we have $|\phi(S)| = 2 \cdot |S| + 1$. Similarly, $\psi(S)$ aims to remove all $\xi$ occurred in $S$. The $(\phi, \psi)$ pair satisfies the property that $\psi(\phi(S)) = S$. We give the following example for a better understanding.

*Example* B.2. Let $\xi := \perp$ for visibility:

$$\phi(\text{Hello}) = \perp\text{H}\perp\text{e}\perp\text{l}\perp\text{l}\perp\text{o}\perp \quad \psi(\perp\text{H}\perp\text{eel}\perp\text{l}\perp\text{o}\perp) = \text{Heello} \quad \psi(\perp\text{H}\perp\text{e}\perp\text{l}\perp\perp\perp\text{o}\perp) = \text{Helo} \quad \psi(\perp\text{H}\perp\text{el}\perp\text{lo}\perp) = \text{Hello} .$$

**Definition B.3** (Replacement operator). Let $S \in \mathcal{S}(\Gamma \cup \{\xi\})$, the integer $i \in [|S|]$ and the character $c$, the replacement operator $\overset{i}{\leftarrow} c$ of the $i^{\text{th}}$ position of $S$ with $c$ is defined as:

$$S \overset{i}{\leftarrow} c := S_{:i-1} \oplus c \oplus S_{i+1:}$$

Thanks to Definition B.3, we are ready to present our attack in Algorithm 1. The advantage of Algorithm 1 resides in attacking a batch of $B$ sentences in parallel, an important feature for efficient adversarial training.

### B.2  Semantic constraints details

In order to follow the semantic constraints of [Chanakya et al., 2024], we constrain the attacks during training and during retrieval and text-to-image generation to not produce new English words. To do so, we employ Algorithm 2 over pairs of sentences $S$ and $S'$ so that $d_{\text{Lev}}(S, S') = 1$. Algorithm 2 returns that the perturbation $S'$ is valid only if it contains less english words than $S$.

### B.3  Training details

All of our text encoders are trained on the first $80,000$ samples of the DataComp-small dataset [Gadre et al., 2023] for 30 epochs with a batch size of 128 sentences. We employ the AdamW optimizer [Kingma and Ba, 2015, Loshchilov and Hutter, 2019], a weight decay of $10^{-4}$, a maximum learning rate of $10^{-5}$ with a linear warmup of $1,400$ steps and cosine decay. For training the robust

**Algorithm 1** LEAF batched attack

1: **Inputs:** Text encoder $\boldsymbol{f_\theta} : \mathcal{S}(\Gamma) \to \mathbb{R}^h$, batch $\{\boldsymbol{S}_i\}_{i=1}^B$, loss function $\mathcal{L}$, radius $k$, number of simultaneous perturbations $\rho$, alphabet $\Gamma$, test character $t$ and flag for semantic constraints Cons.

2: $\hat{\boldsymbol{S}}_i = \boldsymbol{S}_i \; \forall i \in [B]$       ▷ Initialize perturbations with clean sentences.

3: **for** $1, \cdots, k$ **do**

4:      $p_{ij} \sim \text{Unif.}\left([2 \cdot |\hat{\boldsymbol{S}}_i| + 1]\right) \;\; \forall i \in [B] \; \forall j \in [\rho]$      ▷ Sample $\rho$ positions in every sentence.

5:      $\bar{\boldsymbol{S}} = \left\{ \left\{ \psi\left( \phi(\hat{\boldsymbol{S}}_i) \overset{p_{ij}}{\leftarrow} t \right) \right\}_{j=1}^\rho \right\}_{i=1}^B$      ▷ Replace the test character in all $p_{ij}$.

6:      **if** Cons **then**      ▷ Use Algorithm 2 to check if the perturbation is valid, revert otherwise.

7:          $\bar{\boldsymbol{S}}_{ij} = \begin{cases} \bar{\boldsymbol{S}}_{ij} & \text{if } \texttt{valid}(\hat{\boldsymbol{S}}_i, \bar{\boldsymbol{S}}_{ij}) \\ \hat{\boldsymbol{S}}_i & \text{otherwise} \end{cases} \;\; \forall i \in [B] \; \forall j \in [\rho]$

8:      $j_i^\star = \arg\max_{j \in [\rho]} \mathcal{L}\left(\boldsymbol{f_\theta}\left(\bar{\boldsymbol{S}}_{ij}\right)\right)$      ▷ Eval. losses in parallel and get the max.

9:      $c_{ij} \sim \text{Unif.}(\Gamma) \;\; \forall i \in [B] \; \forall j \in [\rho]$      ▷ Sample $\rho$ characters for every sentence.

10:     $\bar{\boldsymbol{S}} = \left\{ \left\{ \psi\left( \phi(\hat{\boldsymbol{S}}_i) \overset{p_{ij_i^\star}}{\leftarrow} c_{ij} \right) \right\}_{j=1}^\rho \right\}_{i=1}^B$      ▷ Replace $c_{ij}$ in the position $p_{ij_i^\star}$.

11:     **if** Cons **then**      ▷ Use Algorithm 2 to check if the perturbation is valid, revert otherwise.

12:         $\bar{\boldsymbol{S}}_{ij} = \begin{cases} \bar{\boldsymbol{S}}_{ij} & \text{if } \texttt{valid}(\hat{\boldsymbol{S}}_i, \bar{\boldsymbol{S}}_{ij}) \\ \hat{\boldsymbol{S}}_i & \text{otherwise} \end{cases} \;\; \forall i \in [B] \; \forall j \in [\rho]$

13:     $l_i^\star = \arg\max_{j \in [\rho]} \mathcal{L}\left(\boldsymbol{f_\theta}\left(\bar{\boldsymbol{S}}_{ij}\right)\right)$      ▷ Eval. losses in parallel and get the max.

14:     $\hat{\boldsymbol{S}}_i = \bar{\boldsymbol{S}}_{il_i^\star} \; \forall i \in [B]$      ▷ Update perturbations.

15: **return** $\left\{ \hat{\boldsymbol{S}}_i \right\}_{i=1}^B$

---

**Algorithm 2** Semantic constraints

1: **Inputs:** Sentence $\boldsymbol{S}$ and perturbation $\boldsymbol{S}'$.

2: $m = |\texttt{words}(\boldsymbol{S})|$

3: $n = |\texttt{words}(\boldsymbol{S}')|$      ▷ We extract English words using NLTK: https://www.nltk.org/

4: **return** $m > n$

---

vision encoder, we adapt the setup of Schlarmann et al. [2024]. Namely, we train on images from ImageNet for 10k steps (instead of 20k, due to compute constraints) with a batch size of 128 for ViT-H/14 and 64 for ViT-g/14. We use weight decay of $10^{-4}$, a maximum learning rate of $10^{-5}$ with a linear warmup of 700 steps and cosine decay. To optimize the inner adversarial objective, we use PGD with 10 steps and set $\epsilon = 2/255$. Our codebase is based on OpenCLIP [Ilharco et al., 2021]. All of our experiments are conducted in a single Nvidia A100 40GB GPU, except for training robust image encoders, where 8 GPUs were employed.

### B.4 Zero-shot text classification

Analogously to how zero-shot image classification is performed in the original CLIP paper [Radford et al., 2021], Qin et al. [2023] encode one image representing each class and compute the similarities with the sentence embedding. Then the predicted class is the one with the highest cosine similarity in the embedding space. In Table 5 we present the images employed for each dataset and label.

### B.5 Text inversion

In order to invert text embeddings, we sample 100 random captions from COCO val2017 and use the optimization method proposed by Wen et al. [2023] with 3000 iterations, learning rate 0.1, and weight decay 0.1.

Table 5: **Images and sentences used for zero-shot text classification.**

| Dataset | Images | | | |
| | Class 1 | Class 2 | Class 3 | Class 4 |
| --- | --- | --- | --- | --- |
| SST-2 / IMDB / Yelp |  |  | NA | NA |
| AG-News |  |  |  |  |
| | Sentences | | | |
| SST-2 / IMDB / Yelp | "Negative Review" | "Positive Review" | NA | NA |
| AG-News | "World News" | "Sports News" | "Business News" | "Science and Technology News" |

Table 6: **Source models employed for finetuning and evaluation.**

| Model | Source |
| --- | --- |
| CLIP-ViT-B-32 | `https://huggingface.co/openai/clip-vit-base-patch32` |
| CLIP-ViT-B-16 | `https://huggingface.co/openai/clip-vit-base-patch16` |
| ViT-L/14 | `https://huggingface.co/openai/clip-vit-large-patch14` |
| FARE | `https://huggingface.co/chs20/fare2-clip` |
| SafeCLIP | `https://huggingface.co/aimagelab/safeclip_vit-l_14` |
| OpenCLIP-ViT-H-14 | `https://huggingface.co/laion/CLIP-ViT-H-14-laion2B-s32B-b79K` |
| OpenCLIP-ViT-g-14 | `https://huggingface.co/laion/CLIP-ViT-g-14-laion2B-s12B-b42K` |
| OpenCLIP-ViT-bigG-14 | `https://huggingface.co/laion/CLIP-ViT-bigG-14-laion2B-39B-b160k` |
| Stable Diffusion v1.5 (SD-1.5) | `https://huggingface.co/stable-diffusion-v1-5/stable-diffusion-v1-5` |
| Stable Diffusion XL base v1.0 (SDXL) | `https://huggingface.co/stabilityai/stable-diffusion-xl-base-1.0` |
| FLUX.1-dev | `https://huggingface.co/black-forest-labs/FLUX.1-dev` |

## B.6 Model checkpoints

In Table 6, we enumerate the external models employed in this work and the sources used for comparison and finetuning.

## C Related work

In this section we cover related work on Adversarial Attacks, Adversarial Training, Robustness of Multimodal Models and text inversion.

**Adversarial Attacks** The vulnerability of deep learning models against adversarial input attacks is well known [Szegedy et al., 2014, Goodfellow et al., 2015] and hast been extensively studied in the vision input domain [Croce and Hein, 2020, Schlarmann and Hein, 2023] and the text input domain, with the most popular attacks employing perturbations in the token-level [Ren et al., 2019, Jin et al., 2020, Li et al., 2019, Garg and Ramakrishnan, 2020, Lee et al., 2022, Ebrahimi et al., 2018, Li et al., 2020, Guo et al., 2021, Hou et al., 2023] and character-level [Belinkov and Bisk, 2018, Ebrahimi et al., 2018, Gao et al., 2018, Pruthi et al., 2019, Yang et al., 2020, Liu et al., 2022, Abad Rocamora et al., 2024].

**Adversarial Training in the text domain.** Adversarial Training [Madry et al., 2018] and its variants [Zhang et al., 2019, Rebuffi et al., 2021, Gowal et al., 2021, Wang et al., 2023, Bartoldson et al., 2024] are the most prominent defense against adversarial examples in the image domain Croce and Hein [2020], Croce et al. [2020].

In the text domain, also variants of adversarial training constitute the best defenses, with most defenses focusing on token-level attacks. Taking advantage of the efficiency of PGD, Miyato et al. [2017] propose solving the inner maximization problem in a $\ell_p$ constrained ball around every token embedding. Zhu et al. [2020] accelerate embedding-level PGD AT and show improvements in clean accuracy. Wang et al. [2021] show improvements in adversarial accuracy by adding an information theoretic regularization term. Deviating from the embedding-based PGD AT paradigm, Dong et al. [2021] use PGD to maximize the loss over a convex combination of synonym embeddings. Then, Hou et al. [2023] find that directly optimizing the inner max in the text space with existing attacks [Jin et al., 2020] significantly boosts the adversarial accuracy against multiple adversarial attacks.

In the character-level, it was initially thought that typo-correctors would suffice as a defense [Pruthi et al., 2019, Jones et al., 2020]. Abad Rocamora et al. [2024] shows that typo-corrector defenses can be easily broken. Additionally Abad Rocamora et al. [2024] show that similarly to the results of [Hou et al., 2023] in the token-level, performing adversarial training with character-level perturbations improved the character-level robustness.

**Robustness of Multimodal Models.** Attacking and defending multimodal models has gained significant interest recently. Mao et al. [2023] propose TeCoA, which performs supervised adversarial fine-tuning on CLIP in order to defend against visual adversarial attacks. In turn, Schlarmann et al. [2024] propose FARE, an unsupervised robust fine-tuning method for vision encoders that preserves downstream performance, e.g. of LMMs that utilize a vision encoder.

**Text inversions.** Morris et al. [2023, 2024] learn models that can invert text embeddings or language model outputs. In contrast, Wen et al. [2023] invert CLIP image embeddings into text via direct optimization. They use the reconstructed text to prompt diffusion models and thereby generate similar images. We use their optimization scheme to invert text embeddings and show that it yields better results when used with our robust models.

# D  Additional experiments

In this section we cover additional experiments not fitting in the main manuscript. First, in Appendix D.1, we analyze the effect adding additional constrains to the adversarial attack. Then, in Appendix D.2 we cover additional experiments in zero-shot classification. In Appendix D.3 we include additional text-to-image generation experiments. I Appendix D.4 we include examples of the sentences reconstructed from their embeddings through embedding inversion. Finally, In Appendix D.6, we perform ablations studying the final losses for different values of $k$ and $\epsilon$, and perform token-level adversarial attacks.

## D.1  On the effect of additional attack constrains for Text-to-image models

In this section, we evaluate the effectiveness of the semantic constraints considered by Chanakya et al. [2024]. In order to avoid including new words with different information in the prompt, Chanakya et al. [2024] constrain the attack to not produce new words in the English vocabulary. To do so, they tokenize the clean and adversarial prompts and check for the appearance of new words in the adversarial prompt based on the NLTK English dictionary [Bird and Loper, 2004]. In order to check for the need of these constraints, we attack SD-1.5 equipped with our robust text encoder at $k = 2$ using Charmer [Abad Rocamora et al., 2024] on the COCO val2017 dataset [Lin et al., 2014]. We then visually explore the adversarial prompts and generated images to look for inconsistencies.

In Table 7 we can observe five examples of unconstrained attacks producing adversarial prompts with significantly different meaning. Since the only constraint is that the Levenshtein distance needs to be $\leq 2$, the attack is able to turn "bear" into "beer", "stop" into "shop", "bananas" into "bandanas" or "wave" into "pave". This results in the diffusion model generating images that correctly adopt these adversarial captions and the adversarial prompts being invalid. If we constrain the attacker to not generate new words, the adversarial prompts preserve the meaning of the original captions up to uncommon words/abvreviations not present in the NLTK dictionary, like "grads" or "smurfs". Overall, we consider the constraints necessary for the text-to-image generation tasks, agreeing with Chanakya et al. [2024].

Table 7: **Examples of problematic attacks in COCO val2017:** If no additional constraints are considered, a single character change can produce semantical changes in the prompt, e.g., "bear" is transformed into "beer". This leads to image generations that are highly dissimilar to the original reference image, but are correct according to the adversarial prompt. The semantic constraints employed by Chanakya et al. [2024] help reducing the amount of new words. Nevertheless, some abbreviations like "grads" or uncommon words like "smurf" still appear after the attack.

| ID | Original caption | Original image | Unconstrained Adversarial caption | Generated image | Constrained [Chanakya et al., 2024] Adversarial caption | Generated image |
|---|---|---|---|---|---|---|
| 285 | A big burly grizzly bear is show with grass in the background. | | A big burly grizzly beer is show with brass in the background. | | A big burly !rizzly bear is show with grads in the background. | |
| 724 | A stop sign is mounted upside-down on it's post. | | A shop sign is mounted up!ide-down on it's post. | | A scop sign is mountedaupside-down on it's post. | |
| 776 | "Three teddy bears, each a different color, snuggling together." | | "Tree teddy beans, each a different color, snuggling together." | | 8hree teddy bears, each a different color, snuggling toge,ther. | |
| 3661 | A bunch of bananas sitting on top of a wooden table. | | A bunch of bandanas sitting on top of aawooden table. | | A bunch of bananas sitti-g on top of a woodenitable. | |
| 6460 | a person riding a surf board on a wave | | a person riding a smurf board on a pave | | a person riding a smurf board on a waze | |

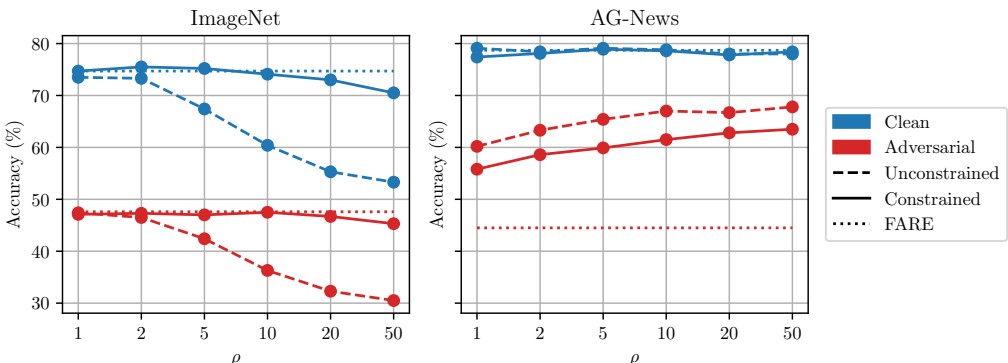

Figure 8: **Hyperparameter effects at** $k = 2$**:** We report the zero-shot clean and adversarial accuracy in both domains (ImageNet and AG-News) with FARE [Schlarmann et al., 2024] as a baseline. For the unconstrained attack, larger values of $\rho$ improve the robustness in the text domain at the cost of significantly degrading the clean and adversarial performance in the image domain. Constraining the attack allows improving the robustness in the text domain with minimal effects on the image domain performance.

## D.2 Zero-shot classification

In this section we include additional datasets for zero-shot image and text classification. We also include a hyperparameter analysis with $k = 2$.

In Fig. 8 we can observe the same experiment as in Section 4.2.2 and Fig. 3 with $k = 2$ instead of $k = 1$. Similarly to the experiments with $k = 1$, increasing $\rho$ leads to a degraded performance in the image domain when no constraints are employed. Including the constraints, allows for increasing

Table 8: **Zero-shot performance for different $k$, $\rho$ and constraints.**

| Semantic Constraints | $k$ | $\rho$ | ImageNet Acc. | ImageNet PGD-20 Acc. ($\epsilon = \frac{2}{255}$) | AG-News Acc. | AG-News Charmer Acc. ($k=1$) |
|---|---|---|---|---|---|---|
| ✗ | 1 | 1 | 74.7 | 46.7 | 78.7 | 57.6 |
| | | 2 | 74.5 | 46.5 | 78.3 | 60.7 |
| | | 5 | 72.0 | 45.4 | 78.7 | 62.9 |
| | | 10 | 70.1 | 43.7 | 78.6 | 64.8 |
| | | 20 | 67.5 | 43.5 | 78.0 | 65.2 |
| | | 50 | 65.5 | 42.0 | 78.2 | 66.3 |
| | 2 | 1 | 73.5 | 47.4 | 79.1 | 60.2 |
| | | 2 | 73.3 | 46.5 | 78.4 | 63.3 |
| | | 5 | 67.4 | 42.4 | 79.1 | 65.4 |
| | | 10 | 60.4 | 36.3 | 78.8 | 67.0 |
| | | 20 | 55.3 | 32.3 | 78.0 | 66.7 |
| | | 50 | 53.3 | 30.5 | 78.0 | 67.8 |
| ✓ | 1 | 1 | 74.7 | 46.9 | 78.2 | 54.4 |
| | | 2 | 74.8 | 47.2 | 77.5 | 56.9 |
| | | 5 | 74.8 | 47.7 | 78.3 | 58.6 |
| | | 10 | 74.8 | 46.3 | 78.3 | 59.9 |
| | | 20 | 73.6 | 46.3 | 78.4 | 60.7 |
| | | 50 | 72.6 | 46.0 | 78.0 | 63.2 |
| | 2 | 1 | 74.7 | 47.1 | 77.4 | 55.8 |
| | | 2 | 75.5 | 47.3 | 78.1 | 58.6 |
| | | 5 | 75.2 | 47.0 | 78.9 | 59.9 |
| | | 10 | 74.1 | 47.5 | 78.6 | 61.5 |
| | | 20 | 73.0 | 46.7 | 77.8 | 62.8 |
| | | 50 | 70.5 | 45.3 | 78.4 | 63.5 |

the robustness in the text domain with less performance degradation. The numbers form Figs. 3 and 8 are available in Table 8.

### D.2.1 Additional experiments on zero-shot image classification

For zero-shot image classification, we measure the clean and robust accuracy on 13 datasets: CalTech101 Griffin et al. [2007], StanfordCars Krause et al. [2013], CIFAR10, CIFAR100 Krizhevsky [2009], DTD Cimpoi et al. [2014], EuroSAT Helber et al. [2019], FGVC Aircrafts Maji et al. [2013], Flowers Nilsback and Zisserman [2008], ImageNet-R Hendrycks et al. [2021], ImageNet-Sketch Wang et al. [2019], PCAM Veeling et al. [2018], OxfordPets Parkhi et al. [2012], and STL-10 Coates et al. [2011]. To measure robustness, we conduct visual attacks as described in Section 4.1, and restrict the evaluation to 1000 random samples on all datasets. We evaluate orginal models and models that employ robust encoders in both domains. Results are reported in Table 9. The robust models maintain much better performance under adversarial attacks, while sacrificing some clean performance.

In Table 10 we report the VTAB [Zhai et al., 2020] averaged performance over the categories *natural*, *specialized*, and *structured*. We observe that in clean evaluation, robust models sacrifice performance on *natural* and *specialized* (a trade-off between clean and robust performance is expected [Tsipras et al., 2019]). On *structured* the behavior is mixed - sometimes even outperforming the non-robust models. In the adversarial evaluation ($\epsilon = 2/255$), we observe that the non-robust models are completely vulnerable, while our robust models maintain much better performance when attacked.

### D.2.2 Additional experiments on zero-shot text classification

In this section, we evaluate the zero-shot clean and adversarial accuracy of our models in additional text classification datasets. We follow the same attack setup as in the AG-News experiments, i.e.,

Table 9: **Zero-shot image classification.** We report the zero-shot image classification performance of original and bimodally robust models.

| | Model | Robust | CalTech101 | Cars | Cifar10 | Cifar100 | DTD | EuroSAT | FGVC | Flowers | ImageNet-r | ImageNet-s | PCAM | Pets | STL10 | Mean |
|---|---|---|---|---|---|---|---|---|---|---|---|---|---|---|---|---|
| Clean | CLIP-ViT-L/14 | ✗ | 82.1 | 77.5 | 95.2 | 68.2 | 55.7 | 63.4 | 28.4 | 79.4 | 86.5 | 48.9 | 53.0 | 93.9 | 98.8 | 71.6 |
| | | ✓ | 81.1 | 71.6 | 92.2 | 68.9 | 44.9 | 28.7 | 24.6 | 69.7 | 83.3 | 47.0 | 59.9 | 91.9 | 98.1 | 66.3 |
| | OpenCLIP-ViT-H/14 | ✗ | 84.4 | 92.2 | 97.5 | 82.8 | 68.7 | 72.5 | 42.4 | 80.2 | 88.4 | 56.1 | 54.9 | 95.1 | 98.1 | 77.9 |
| | | ✓ | 83.8 | 89.8 | 93.3 | 69.7 | 61.1 | 34.4 | 35.8 | 73.4 | 85.7 | 52.9 | 50.4 | 94.0 | 97.2 | 70.9 |
| | OpenCLIP-ViT-g/14 | ✗ | 84.3 | 92.1 | 97.7 | 84.0 | 68.8 | 65.6 | 36.4 | 78.1 | 88.2 | 55.5 | 55.6 | 95.2 | 98.2 | 76.9 |
| | | ✓ | 83.1 | 88.4 | 91.7 | 67.3 | 58.1 | 29.0 | 30.7 | 71.2 | 84.9 | 52.0 | 52.5 | 92.5 | 96.2 | 69.0 |
| $\epsilon = 2/255$ | CLIP-ViT-L/14 | ✗ | 0.0 | 0.0 | 0.0 | 0.0 | 0.0 | 0.0 | 0.0 | 0.0 | 0.0 | 0.0 | 0.0 | 0.0 | 0.0 | 0.0 |
| | | ✓ | 70.5 | 27.8 | 65.6 | 34.2 | 25.3 | 11.6 | 6.0 | 33.8 | 55.5 | 26.4 | 22.1 | 69.0 | 89.7 | 41.3 |
| | OpenCLIP-ViT-H/14 | ✗ | 0.0 | 0.0 | 0.3 | 0.2 | 0.0 | 0.0 | 0.0 | 0.0 | 0.0 | 0.0 | 0.0 | 0.0 | 0.0 | 0.0 |
| | | ✓ | 70.7 | 55.6 | 65.0 | 38.4 | 32.5 | 7.7 | 5.8 | 39.5 | 58.3 | 31.0 | 37.9 | 66.0 | 87.9 | 45.9 |
| | OpenCLIP-ViT-g/14 | ✗ | 0.0 | 0.0 | 0.1 | 0.2 | 0.0 | 0.0 | 0.0 | 0.0 | 0.0 | 0.0 | 0.0 | 0.0 | 0.0 | 0.0 |
| | | ✓ | 71.3 | 52.1 | 62.6 | 34.0 | 28.5 | 4.7 | 4.0 | 34.2 | 53.3 | 28.6 | 26.5 | 57.5 | 84.7 | 41.7 |

Table 10: **VTAB zero-shot image classification.** We report the zero-shot image classification performance of original and bimodally robust models on VTAB Zhai et al. [2020].

| | Model | Robust | Natural | Specialized | Structured |
|---|---|---|---|---|---|
| Clean | ViT-L/14 | ✗ | 74.4 | 63.5 | 11.9 |
| | | ✓ | 68.5 | 41.9 | 13.3 |
| | ViT-H/14 | ✗ | 78.7 | 57.0 | 11.7 |
| | | ✓ | 74.8 | 45.6 | 11.8 |
| | ViT-g/14 | ✗ | 79.5 | 62.9 | 12.5 |
| | | ✓ | 72.4 | 51.4 | 11.4 |
| $\epsilon = 2/255$ | ViT-L/14 | ✗ | 0.0 | 0.0 | 0.0 |
| | | ✓ | 42.4 | 10.6 | 3.9 |
| | ViT-H/14 | ✗ | 0.1 | 0.0 | 0.0 |
| | | ✓ | 44.9 | 14.6 | 3.6 |
| | ViT-g/14 | ✗ | 0.0 | 0.0 | 0.0 |
| | | ✗ | 41.0 | 9.5 | 1.9 |

we employ Charmer-20 at $k = 1$ without semantic constraints to evaluate the performance on SST-2 [Socher et al., 2013], IMDB [Maas et al., 2011] and Yelp [Yelp, 2015, Zhang et al., 2015].

In Fig. 9 we report the zero-shot adversarial accuracy already reported in Fig. 4, with the addition of SafeCLIP [Poppi et al., 2024]. SafeCLIP obtains a considerably lower clean and adversarial accuracy in comparison to the other CLIP variants.

In Table 11 we can observe that similarly to the AG-News results in Table 2, the models with robust text encoders achieve higher adversarial accuracy in the text domain, with improvements of more than 9.9 robust accuracy points for all models and datasets.

In Table 12, we present the clean and adversarial zero-shot accuracy when employing only the text encoder for the ViT-L/14 models. For that, we encode on sentence per label instead of one image per label as done in the main text. See Table 5 for more details on the sentences employed for the labels. We can observe that the adversarial accuracy is larger after adversarial finetuning with LEAF. Nevertheless, the clean and adversarial performance are worse when doing text-encoder-only zero-shot classification, e.g., a clean accuracy in AG-News with ViT-L/14 of 74.4 when using images as labels (Table 2) v.s. 54.8 when using sentences as labels.

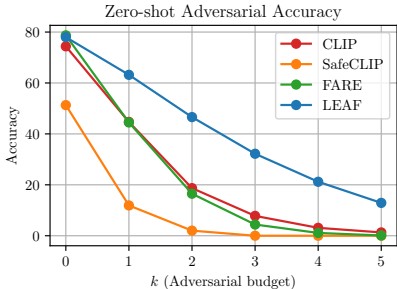

Figure 9: **Larger perturbations:** We evaluate the adversarial accuracy in AG-News for $k \in \{1, 2, 3, 4, 5\}$ in the ViT-L/14 scale. Our model (LEAF) obtains the highest adversarial accuracy at all values of the distance bound $k$.

Table 11: **Zero-shot text classification.** We report the zero-shot text classification performance of original and bimodally robust models.

|  | Model | Robust | SST-2 | IMDB | Yelp |
|---|---|---|---|---|---|
| Clean | CLIP-ViT-L/14 | ✗ | 71.2 | 61.6 | 80.9 |
|  |  | ✓ | 71.9 | 61.4 | 82.0 |
|  | OpenCLIP-ViT-H/14 | ✗ | 61.6 | 57.5 | 73.7 |
|  |  | ✓ | 58.4 | 53.2 | 72.6 |
|  | OpenCLIP-ViT-g/14 | ✗ | 57.8 | 56.8 | 71.9 |
|  |  | ✓ | 56.0 | 54.0 | 71.1 |
| $k = 1$ | CLIP-ViT-L/14 | ✗ | 6.8 | 13.7 | 21.0 |
|  |  | ✓ | 23.2 | 31.0 | 43.8 |
|  | OpenCLIP-ViT-H/14 | ✗ | 16.2 | 31.1 | 22.1 |
|  |  | ✓ | 36.4 | 43.9 | 40.8 |
|  | OpenCLIP-ViT-g/14 | ✗ | 21.4 | 31.4 | 26.0 |
|  |  | ✓ | 34.2 | 41.3 | 39.4 |

Table 12: **Text-encoder-only zero-shot text classification:** We report the clean and adversarial zero shot accuracy at $k = 1$ employing only text-encoders. The adversarial accuracy improves after adversarial finetuning with LEAF. Nevertheless, employing only the text encoder provides worse clean and adversarial performance than employing images as labels as Qin et al. [2023].

| | AG-News | | SST-2 | | IMDB | | Yelp | |
|---|---|---|---|---|---|---|---|---|
| Robust | Acc. | Adv. | Acc. | Adv. | Acc. | Adv. | Acc. | Adv. |
| ✗ | 54.8 | 17.9 | 60.3 | 3.2 | 54.0 | 24.9 | 59.9 | 29.5 |
| ✓ | 53.5 | 34.7 | 58.9 | 24.1 | 51.5 | 44.9 | 56.7 | 47.5 |

### D.3 Additional experiments in text-to-image models

In this section, we provide additional experiments and examples for the text-to-image generation task. In Tables 13 and 14 we present the generation results in SD-1.5 and SDXL in the MS-COCO dataset and the first 5.000 images of the Flickr30k dataset. We measure the CLIPScore between the original caption and the generated image (T-I), the CLIPScore between the original image and the generated one (I-I), the attack objective (Eq. (2)) and for SD-1.5, the percentage of generated images triggering the NSFW filter (NSFW %). We can observe that the text encoders finetuned with LEAF, provide a higher generation quality for $k > 1$ according to all generation metrics. Surprisingly, for $k = 2$ and $k = 4$ in the MS-COCO dataset, our text encoders triggered the NSFW filter less frequently than SafeCLIP [Poppi et al., 2024], which is specifically designed to avoid generating NSFW content.

In Tables 15 to 18 we present examples of the attacks on the first 10 samples of each dataset for both SD-1.5 and SDXL at $k = 2$. We can observe, that our text encoders provide qualitatively better images. The models with the original text encoders, provide images unrelated to the original image and caption more often than the models employing our text encoders.

In Table 19 we include the generation results with FLUX.1-dev [Black Forest Labs et al., 2025]. Since FLUX.1-dev employs CLIP ViT-L/14 and FLAN-T5 XXL [Chung et al., 2022] as text encoders, the model can only be benefited from our approach by replacing the CLIP text encoder with our LEAF counterpart. Similarly, we only attack the CLIP / LEAF text encoders and assume no access to FLAN-T5 XXL. Due to the high resolution of the FLUX.1-dev generations ($1024 \times 1024$), we restrict the evaluation of FLUX.1-dev to the first 100 images in the MS-COCO validation set.

#### D.3.1 Transfer attacks on text-to-image models

In this section we evaluate the performance of transfer attacks on SD-1.5 with CLIP and LEAF as either the source model where the attack is optimized or the target model used for the image generation. In Table 20 we can observe that, as expected, when the source is equal to the target, the generated image quality is degraded the most. Our text encoder improves the generation quality in all cases except when the source is LEAF and $k = 1$, where CLIP obtains 0.04 more CLIPScore T2I score points than LEAF in this advantageous setup.

#### D.3.2 Preliminary study of typographic attacks

In this section we evaluate how our text encoder preserves the image quality under typographic prompts, i.e., prompts where characters have been changed for visually similar ones. To do so, we emply SD-1.5 and replace every "i" for a "1", every "e" for a "3", every "o" for a "0" and every "a" for an "@" in the first 100 prompts in the MS-COCO dataset. As an example, the first COCO caption turns into "A w0m@n st@nds 1n th3 d1n1ng @r3@ @t th3 t@bl3."

In Table 21, we can observe that while the image generation quality with both encoders is quite low, using LEAF provides an improvement of 0.62 points in CLIPScore T2I and 2.77 in CLIPScore I2I.

### D.4 Embedding inversion examples

In Tables 22 and 23 we present examples from the embedding-to-text reconstructions results performed in Section 4.6.

### D.5 Additional retrieval experiments

For $1,000$ validation set queries, the attack explained in the main part maximizes the similarity between the test query and a target string using different variants of the Charmer attack. In Table 24, we show the individual attack results across 3 target strings for differently trained LEAF models. One sees that on increasing training $\rho$, the robustness goes up with a slight decay in the clean retrieval performance. This trade-off is similar to the one seen for classification tasks in Fig. 3.

In Fig. 10, we visualize the top-3 retrieved images for the original and the perturbed queries. Although in some cases the non robust model retrieves a relevant query, the top-1 retrieved image is always different for clean and perturbed queries. However, the robust model always preserves the original top-1 retrieved image showing its robustness to such character perturbed queries.

Table 13: **Text-to-image generation results on MS-COCO:** SD-1.5 and SDXL are evaluated over the full 5000 images in the valudation set. FLUX.1-dev is evaluated over the first 100 images due to the high resolution of the generated images.

| Pipeline | k | Text encoder | $\text{Sim}(f_\theta(S), f_\theta(S'))$ | CLIPScore T2I | CLIPScore I2I | NSFW (%) |
|---|---|---|---|---|---|---|
| SD-1.5 | 0 | CLIP | - | $\mathbf{31.50}_{(\pm 2.87)}$ | $\mathbf{73.31}_{(\pm 10.21)}$ | 0.64 |
| | | SafeCLIP | | $30.96_{(\pm 2.93)}$ | $73.27_{(\pm 10.08)}$ | **0.44** |
| | | LEAF | | $31.00_{(\pm 2.94)}$ | $73.06_{(\pm 10.12)}$ | 0.46 |
| | 1 | CLIP | $55.85_{(\pm 8.66)}$ | $27.53_{(\pm 4.52)}$ | $65.38_{(\pm 12.71)}$ | 0.96 |
| | | SafeCLIP | $71.62_{(\pm 8.32)}$ | $27.43_{(\pm 4.09)}$ | $66.90_{(\pm 11.56)}$ | **0.48** |
| | | LEAF | $\mathbf{86.58}_{(\pm 4.84)}$ | $\mathbf{27.96}_{(\pm 3.48)}$ | $\mathbf{68.01}_{(\pm 11.17)}$ | 0.50 |
| | 2 | CLIP | $33.18_{(\pm 9.29)}$ | $22.96_{(\pm 5.79)}$ | $57.21_{(\pm 13.90)}$ | 2.16 |
| | | SafeCLIP | $50.87_{(\pm 10.34)}$ | $23.75_{(\pm 5.02)}$ | $61.02_{(\pm 12.06)}$ | 1.08 |
| | | LEAF | $\mathbf{73.15}_{(\pm 7.45)}$ | $\mathbf{25.23}_{(\pm 4.36)}$ | $\mathbf{63.40}_{(\pm 11.95)}$ | **0.62** |
| | 3 | CLIP | $20.38_{(\pm 8.93)}$ | $19.45_{(\pm 5.86)}$ | $51.55_{(\pm 13.40)}$ | 2.52 |
| | | SafeCLIP | $35.93_{(\pm 11.06)}$ | $20.41_{(\pm 5.61)}$ | $55.98_{(\pm 12.07)}$ | **1.10** |
| | | LEAF | $\mathbf{60.00}_{(\pm 9.07)}$ | $\mathbf{22.59}_{(\pm 5.16)}$ | $\mathbf{59.02}_{(\pm 12.19)}$ | 1.26 |
| | 4 | CLIP | $12.83_{(\pm 8.80)}$ | $17.42_{(\pm 5.68)}$ | $48.34_{(\pm 12.66)}$ | 2.70 |
| | | SafeCLIP | $26.05_{(\pm 11.04)}$ | $17.94_{(\pm 5.57)}$ | $52.31_{(\pm 11.57)}$ | 1.56 |
| | | LEAF | $\mathbf{49.35}_{(\pm 9.55)}$ | $\mathbf{20.25}_{(\pm 5.44)}$ | $\mathbf{55.36}_{(\pm 12.33)}$ | **1.44** |
| SDXL | 0 | CLIP + OpenCLIP | - | $\mathbf{31.90}_{(\pm 2.84)}$ | $\mathbf{71.87}_{(\pm 10.58)}$ | - |
| | | 2×LEAF | | $31.80_{(\pm 2.86)}$ | $71.78_{(\pm 10.60)}$ | |
| | 1 | CLIP + OpenCLIP | $67.65_{(\pm 7.46)}$ | $28.33_{(\pm 4.11)}$ | $64.45_{(\pm 12.25)}$ | |
| | | 2×LEAF | $\mathbf{88.15}_{(\pm 4.44)}$ | $\mathbf{29.37}_{(\pm 3.46)}$ | $\mathbf{67.25}_{(\pm 11.54)}$ | |
| | 2 | CLIP + OpenCLIP | $47.58_{(\pm 8.74)}$ | $24.65_{(\pm 5.25)}$ | $57.97_{(\pm 12.89)}$ | |
| | | 2×LEAF | $\mathbf{76.49}_{(\pm 7.12)}$ | $\mathbf{27.14}_{(\pm 4.33)}$ | $\mathbf{63.27}_{(\pm 12.19)}$ | |
| | 3 | CLIP + OpenCLIP | $34.22_{(\pm 8.90)}$ | $21.45_{(\pm 5.70)}$ | $53.37_{(\pm 12.78)}$ | |
| | | 2×LEAF | $\mathbf{64.62}_{(\pm 9.24)}$ | $\mathbf{24.69}_{(\pm 5.16)}$ | $\mathbf{59.38}_{(\pm 12.66)}$ | |
| | 4 | CLIP + OpenCLIP | $25.93_{(\pm 8.74)}$ | $19.07_{(\pm 5.60)}$ | $49.92_{(\pm 12.21)}$ | |
| | | 2×LEAF | $\mathbf{54.08}_{(\pm 10.22)}$ | $\mathbf{22.45}_{(\pm 5.67)}$ | $\mathbf{55.70}_{(\pm 12.85)}$ | |
| FLUX.1-dev | 0 | CLIP + FLAN-T5 XXL | - | $\mathbf{30.56}_{(\pm 2.86)}$ | $\mathbf{71.19}_{(\pm 12.13)}$ | - |
| | | LEAF + FLAN-T5 XXL | | $30.55_{(\pm 2.90)}$ | $71.18_{(\pm 12.83)}$ | |
| | 1 | CLIP + FLAN-T5 XXL | $57.86_{(\pm 8.70)}$ | $\mathbf{29.14}_{(\pm 3.76)}$ | $68.09_{(\pm 12.82)}$ | |
| | | LEAF + FLAN-T5 XXL | $\mathbf{87.07}_{(\pm 4.52)}$ | $28.90_{(\pm 3.60)}$ | $\mathbf{68.79}_{(\pm 12.91)}$ | |
| | 2 | CLIP + FLAN-T5 XXL | $35.04_{(\pm 8.87)}$ | $27.03_{(\pm 5.20)}$ | $63.60_{(\pm 13.51)}$ | |
| | | LEAF + FLAN-T5 XXL | $\mathbf{73.70}_{(\pm 6.90)}$ | $\mathbf{27.38}_{(\pm 4.09)}$ | $\mathbf{65.66}_{(\pm 13.01)}$ | |
| | 3 | CLIP + FLAN-T5 XXL | $21.84_{(\pm 7.78)}$ | $24.47_{(\pm 6.00)}$ | $59.40_{(\pm 14.09)}$ | |
| | | LEAF + FLAN-T5 XXL | $\mathbf{59.83}_{(\pm 9.23)}$ | $\mathbf{25.71}_{(\pm 5.16)}$ | $\mathbf{62.11}_{(\pm 13.84)}$ | |
| | 4 | CLIP + FLAN-T5 XXL | $14.79_{(\pm 7.10)}$ | $22.72_{(\pm 6.11)}$ | $57.68_{(\pm 14.33)}$ | |
| | | LEAF + FLAN-T5 XXL | $\mathbf{49.57}_{(\pm 9.86)}$ | $\mathbf{23.51}_{(\pm 5.98)}$ | $\mathbf{59.59}_{(\pm 15.27)}$ | |

Table 14: **Text-to-image generation results on Flickr30k:**

| Pipeline | k | Text encoder | $\text{Sim}(f_{\boldsymbol{\theta}}(\boldsymbol{S}), f_{\boldsymbol{\theta}}(\boldsymbol{S'}))$ | CLIPScore T2I | CLIPScore I2I | NSFW (%) |
|---|---|---|---|---|---|---|
| SD-1.5 | 0 | CLIP | | $\mathbf{33.27}_{(\pm 3.21)}$ | $\mathbf{71.27}_{(\pm 10.20)}$ | 0.42 |
| | | SafeCLIP | - | $32.16_{(\pm 3.35)}$ | $70.20_{(\pm 10.25)}$ | 0.42 |
| | | LEAF | | $32.63_{(\pm 3.17)}$ | $70.73_{(\pm 10.23)}$ | **0.26** |
| | 1 | CLIP | $63.48_{(\pm 9.01)}$ | $\mathbf{30.72}_{(\pm 4.16)}$ | $66.43_{(\pm 11.25)}$ | 0.84 |
| | | SafeCLIP | $77.31_{(\pm 7.11)}$ | $29.32_{(\pm 4.19)}$ | $65.68_{(\pm 10.85)}$ | 0.92 |
| | | LEAF | $\mathbf{89.80}_{(\pm 3.89)}$ | $30.37_{(\pm 3.56)}$ | $\mathbf{67.54}_{(\pm 10.56)}$ | **0.66** |
| | 2 | CLIP | $42.37_{(\pm 10.21)}$ | $27.71_{(\pm 5.18)}$ | $61.28_{(\pm 12.18)}$ | 1.28 |
| | | SafeCLIP | $59.79_{(\pm 9.63)}$ | $26.24_{(\pm 4.72)}$ | $61.66_{(\pm 11.12)}$ | 0.87 |
| | | LEAF | $\mathbf{79.28}_{(\pm 6.55)}$ | $\mathbf{28.43}_{(\pm 4.05)}$ | $\mathbf{64.66}_{(\pm 10.80)}$ | **0.68** |
| SDXL | 0 | CLIP + OpenCLIP | - | $\mathbf{33.85}_{(\pm 3.24)}$ | $\mathbf{69.07}_{(\pm 10.54)}$ | |
| | | 2×LEAF | | $33.82_{(\pm 3.22)}$ | $69.06_{(\pm 10.50)}$ | |
| | 1 | CLIP + OpenCLIP | $75.15_{(\pm 6.33)}$ | $31.24_{(\pm 4.00)}$ | $64.03_{(\pm 11.23)}$ | - |
| | | 2×LEAF | $\mathbf{91.32}_{(\pm 3.40)}$ | $\mathbf{31.63}_{(\pm 3.54)}$ | $\mathbf{65.87}_{(\pm 10.89)}$ | |
| | 2 | CLIP + OpenCLIP | $58.02_{(\pm 8.49)}$ | $28.30_{(\pm 4.81)}$ | $59.09_{(\pm 11.47)}$ | |
| | | 2×LEAF | $\mathbf{82.82}_{(\pm 5.84)}$ | $\mathbf{29.83}_{(\pm 4.09)}$ | $\mathbf{63.03}_{(\pm 11.15)}$ | |

Table 15: **Attack examples on MS-COCO with SD-1.5 at** $k = 2$**:** The color borders indicate null, partial and total matching to the original image caption. The model with the original text encoder provides images involving a footballer, a lizard or a gun, when prompted about a bear, a women skiing or a group of people respectively. With our text encoders, the generation does not drift in topic so much.

| ID | Original caption | Original image | Original | | SafeCLIP | | LEAF | |
|---|---|---|---|---|---|---|---|---|
| | | | Adversarial caption | Generated image | Adversarial caption | Generated image | Adversarial caption | Generated image |
| 139 | A woman stands in the dining area at the table. | | A woman stan3s in the dining area at the table- | | A woma2 stands in the cining area at the table. | | Avwomanastands in the dining area at the table. | |
| 285 | A big burly grizzly bear is show with grass in the background. | | A big burly grie zly bear is show with g?rass in the background. | | A big burly gr#izzly bearvis show with grass in the back-ground. | | A big burly .rizzly bear is show with @rass in the background. | |
| 632 | Bedroom scene with a bookcase, blue comforter and window. | | Bedr=oom scene with a bookcase, blue comfor#ter and window. | | Bedroom scene with a @ookcase, bl#ue com-forter and window. | | Bedroomascene with a kookcase, blue comforter and window. | |
| 724 | A stop sign is mounted upside-down on it's post. | | A stop si$gn is mounted upsixde-down on it's post. | | A stox sign is mounted upside-down on it's pos$. | | A stopssign is mounted upside-downton it's post. | |
| 776 | Three teddy bears, each a different color, snuggling together. | | Thr|e teddy sears, each= a different color, snuggling together. | | Thr|ee teddy bears, eac= a different color, snuggling together. | | 9hree teddy bears, each a different color, snuggling toge,ther. | |
| 785 | A woman posing for the camera standing on skis. | | A woma6n posing for the camera standing on >kis. | | A woma6 posing for the camera stand-ing onuskis. | | A -oman posing for the camera stand-ing onoskis. | |
| 802 | A kitchen with a refrigerator, stove and oven with cabinets. | | A kit>chen with a re-frigerator, stove and oven withmcabinets. | | A kiltchen with a refr#igerator, stove and oven with cabinets. | | Aqkitchen withra refrig-erator, stove and oven with cabinets. | |
| 872 | A couple of baseball player stand-ing on a field. | | A couple of basmball player stand-ing on a fi#eld. | | A cozuple of basebalm player stand-ing on a field. | | A coupl. of baseball player stand-ing on a ˆield. | |
| 885 | a male tennis player in white shorts is playing tennis | | a male ten=is player in white shor?ts is play-ing tennis | | a male ten-nis player in wh.ite )horts is playing tennis | | aimale tennis playercin white shorts is playing tennis | |
| 1000 | The people are posing for a group photo. | | The pzople are posing for a group ph6oto. | | The people are posi?ng for a gr1oup photo. | | The people are posing forza group bhoto. | |

Table 16: **Attack examples on MS-COCO with SDXL at $k = 2$:**

| ID | Original caption | Original image | Original Adversarial caption | Original Generated image | LEAF Adversarial caption | LEAF Generated image |
|---|---|---|---|---|---|---|
| 139 | A woman stands in the dining area at the table. | | A woma8 stands in the jining area at the table. | | 3 woman'stands in the dining area at the table. | |
| 285 | A big burly grizzly bear is show with grass in the background. | | A big burly gr1izzly bear is show with @rass in the background. | | A big burly !rizzly bear is show with krass in the background. | |
| 632 | Bedroom scene with a bookcase, blue comforter and window. | | Bedroom sc]ene with a zookcase, blue comforter and window. | | Bedroom scene with a cookcase, blue cosmforter and window. | |
| 724 | A stop sign is mounted upside-down on it's post. | | A stop gign is mountedpupside-down on it's post. | | A 3top sign is mounted upside-downton it's post. | |
| 776 | Three teddy bears, each a different color, snuggling together. | | Thr:ee teddy bears, each a different color, snuggling toge—ther. | | ahree teddy bears, each a different color, snuggling toge,ther. | |
| 785 | A woman posing for the camera standing on skis. | | A woma: posing for the camera standing ontskis. | | A -oman posing for the camera standing onoskis. | |
| 802 | A kitchen with a refrigerator, stove and oven with cabinets. | | A ki:chen with a refr@igerator, stove and oven with cabinets. | | Aqkitchen withra refrigerator, stove and oven with cabinets. | |
| 872 | A couple of baseball player standing on a field. | | A couple of basebill player standing on a #ield. | | A coupll of baseball player standing on a qield. | |
| 885 | a male tennis player in white shorts is playing tennis | | a male tennis pl*ayer in white #horts is playing tennis | | aemale tennis playerein white shorts is playing tennis | |
| 1000 | The people are posing for a group photo. | | The neople are posing for a group |hoto. | | The peoplecare posing forza group photo. | |

#### D.5.1 Bimodal attacks in text-to-image retrieval

Building on top of text-modality robustness for text-to-image retrieval from the main part, we now assess the robustness to bimodal attacks for both the image and text modalities for $1k$ samples of the MS-COCO test set. The evaluation starts from the known baseline ($k = 1$ text perturbations) from Table 3 and applies an untargeted adversarial attack to the images. We use APGD [Croce and Hein, 2020] for 100 iterations with small $\ell_\infty$ perturbation radii of $2/255$ and $4/255$. This perturbation is designed to maximize the distance between the original and perturbed image embeddings, thereby disrupting the model's ability to retrieve the correct text. This attack protocol, is similar to CoAttack [Zhang et al., 2022], where the text attack follows the image attack.

The results in Table 25 highlight the superior resilience of the LEAF-trained models. For the critical recall@1 metric, LEAF improved retrieval performance by nearly 7% over the baseline across both perturbation radii. Importantly, this significant gain in robustness did not come at the cost of clean performance (performance on clean data), as indicated by the 'clean' column results. This finding strongly underscores the importance of dual modality robustness: the ability to maintain high performance despite adversarial attacks on either the image or text data, making LEAF the most robust solution in this challenging setup.

Table 17: **Attack examples on Flickr30k with SD-1.5 at** $k = 2$**:**

| ID | Original caption | Original image | Original | | SafeCLIP | | LEAF | |
|---|---|---|---|---|---|---|---|---|
| | | | Adversarial caption | Generated image | Adversarial caption | Generated image | Adversarial caption | Generated image |
| 1000092795 | Two young guys with shaggy hair look at their hands while hanging out in the yard . |  | Two young guys with shagg) hair look at their hands while hanging out in the #ard . |  | Two young guys with shaggychair zook at their hands while hanging out in the yard . |  | Twt young guys with shaggy hair look at their hands while hanging out in the mard . |  |
| 10002456 | Several men in hard hats are operating a giant pulley system . |  | Severa= men in hard hats are operat{ng a giant pulley system . |  | Several menxin hardghats are operating a giant pulley system . |  | Severalumen in harz hats are operating a giant pulley system . |  |
| 1000268201 | A child in a pink dress is climbing up a set of stairs in an entry way . |  | A ch:ld in a pink dr_ss is climbing up a set of stairs in an entry way . |  | A ch&ld in aapink dress is climbing up a set of stairs in an entry way . |  | A 8hild in a pink dr·ss is climbing up a set of stairs in an entry way . |  |
| 1000344755 | Someone in a blue shirt and hat is standing on stair and leaning against a window . |  | Someone in a bl2ue shirt and hat is standing on stair and leaning against a windo& . |  | Someone in a bl4ue shirt and hat is standing on @tair and leaning against a window . |  | Someone in a blue shirt andphat is standing on &tair and leaning against a window . |  |
| 1000366164 | Two men , one in a gray shirt , one in a black shirt , standing near a stove . |  | +wo men , one in a gray shirt , one in a black shirt , standing near a @tove . |  | 7wo men , one in a gray shirt , one in a black shirt , standing near a 1tove . |  | cwo men , one in a gray shirt , one in a black shirt , standing near a .tove . |  |
| 1000523639 | Two people in the photo are playing the guitar and the other is poking at him . |  | Two people in the photo are playing the guita] and the other is poki.ng at him . |  | 7Two pefople in the photo are playing the guitar and the other is poking at him . |  | Two people in the photocare playing the suitar and the other is poking at him . |  |
| 1000919630 | A man sits in a chair while holding a large stuffed animal of a lion . |  | A man sits in a chair whil$ holding a large stu!ffed animal of a lion . |  | A manpsits in a ch2ir while holding a large stuffed animal of a lion . |  | A man sits in a chair while holding a large stunffed animal of a liox . |  |
| 10010052 | A girl is on rollerskates talking on her cellphone standing in a parking lot . |  | A gorl is on rollerskates talking on her cellphone standing in a parki{ng lot . |  | A gir% is on rollerskates talking on herwcellphone standing in a parking lot . |  | Adgirl is on rollerskates talking on her cellphone standing in a parkingelot . |  |
| 1001465944 | An asian man wearing a black suit stands near a dark-haired woman and a brown-haired woman . |  | An asian man wearing a bl#ack sui@ stands near a dark-haired woman and a brown-haired woman . |  | An asian man wearing a blauck sui! stands near a dark-haired woman and a brown-haired woman . |  | Ankasian man wearing a black suit stands near a dark-haired woman !nd a brown-haired woman . |  |
| 1001545525 | Two men in Germany jumping over a rail at the same time without shirts . |  | Twy men in Germany jumping over aarail at the same time without shirts . |  | Two men in Germany jumping over a raij at the same time withouk shirts . |  | cwo men in Germany jumping over a rail at the same time !ithout shirts . |  |

Table 18: **Attack examples on Flickr30k with SDXL at $k = 2$:**

| ID | Original caption | Original image | Original | | LEAF | |
| | | | Adversarial caption | Generated image | Adversarial caption | Generated image |
|---|---|---|---|---|---|---|
| 1000092795 | Two young guys with shaggy hair look at their hands while hanging out in the yard . |  | Two young guys with shaggychair look at their hands while hanging out in the \|ard . |  | Two young guys with shaggychair look at their hands while hanging out in the mard . |  |
| 10002456 | Several men in hard hats are operating a giant pulley system . |  | Several men in $ard hats are operating a giant !ulley system . |  | Several men in hardchats are operating a giant sulley system . |  |
| 1000268201 | A child in a pink dress is climbing up a set of stairs in an entry way . |  | A ch\|ld in a pink dr_ss is climbing up a set of stairs in an entry way . |  | A chwild in a pink dress is climbing up a set of stairs in ankentry way . |  |
| 1000344755 | Someone in a blue shirt and hat is standing on stair and leaning against a window . |  | Someone in a bl2ue shirt and hat is standing on stair and leaning against a :indow . |  | Someone in a blue shirt andwhat is standing on &tair and leaning against a window . |  |
| 1000366164 | Two men , one in a gray shirt , one in a black shirt , standing near a stove . |  | Twomen , one in a gray shirt , one in a black shirt , standing near a @tove . |  | Twomen , one in a gray shirt , one in a black shirt , standing near a ptove . |  |
| 1000523639 | Two people in the photo are playing the guitar and the other is poking at him . |  | Two people in the ph?oto are playing the gu#itar and the other is poking at him . |  | Two people in the photo are playing the suitarmand the other is poking at him . |  |
| 1000919630 | A man sits in a chair while holding a large stuffed animal of a lion . |  | A man sits in a ch5ir while holding a large stu!ffed animal of a lion . |  | A man sits in a chair while holding a large stuffe. animal of aklion . |  |
| 10010052 | A girl is on rollerskates talking on her cellphone standing in a parking lot . |  | A girl is on rollerskates talking on her cellphone standing in a parki{ngblot . |  | A girl is on rollerskatesstalking on her cellphone standing in a parkingslot . |  |
| 1001465944 | An asian man wearing a black suit stands near a dark-haired woman and a brown-haired woman . |  | Axn asian man wearing a black suit stands near a dark-haired woman #nd a brown-haired woman . |  | Axn asian man wearing a black suit stands near a dark-haired woman anz a brown-haired woman . |  |
| 1001545525 | Two men in Germany jumping over a rail at the same time without shirts . |  | Two men in Germany jumping over a rai7 at the same time ?ithout shirts . |  | cwo men in Germany jumping over a rail at the same time ?ithout shirts . |  |

Table 19: **Attack examples on MS-COCO with FLUX.1-dev at $k = 2$:**

| ID | Original caption | Original image | Original Adversarial caption | Generated image | LEAF Adversarial caption | Generated image |
|---|---|---|---|---|---|---|
| 139 | A woman stands in the dining area at the table. | | A woma@ stands in the xining area at the table. | | Avwomanastands in the dining area at the table. | |
| 285 | A big burly grizzly bear is show with grass in the background. | | A big burly griezly bear is show with ?rass in the background. | | A big burly .rizzly bear is show with @rass in the background. | |
| 632 | Bedroom scene with a bookcase, blue comforter and window. | | Bedr=oom scene with a bookcase, blue comfor#ter and window. | | Bedroomascene with a kookcase, blue comforter and window. | |
| 724 | A stop sign is mounted upside-down on it's post. | | A stop $ign is mounted upside-down on fit's post. | | A stopssign is mounted upside-downton it's post. | |
| 776 | Three teddy bears, each a different color, snuggling together. | | +hree teddy bears, each a different color, snuggling @ogether. | | 9hree teddy bears, each a different color, snuggling toge,ther. | |
| 785 | A woman posing for the camera standing on skis. | | A woma7 posing for the camera standing onoskis. | | A -oman posing for the camera standing onoskis. | |
| 802 | A kitchen with a refrigerator, stove and oven with cabinets. | | A ki=chen with a refrigeratoa, stove and oven with cabinets. | | Aqkitchen withra refrigerator, stove and oven with cabinets. | |
| 872 | A couple of baseball player standing on a field. | | A couple of $aseball player standing on a #ield. | | A coupl. of baseball player standing on a ∧ield. | |
| 885 | a male tennis player in white shorts is playing tennis | | a malec tennis pl*ayer in white shorts is playing tennis | | aimale tennis playerein white shorts is playing tennis | |
| 1000 | The people are posing for a group photo. | | The neople are posing for a group ph?oto. | | The people are posing forza group bhoto. | |

Table 20: **Transfer attacks in SD-1.5:** Columns represent the source text encoder, where the attack is optimized, and rows the target text encoder, where the attack is evaluated. LEAF obtains the highest CLIPScores for every setup except the CLIPScore T2I at $k = 1$ with LEAF as a source model.

| k | Target \Source | CLIPScore T2I | | CLIPScore I2I | |
|---|---|---|---|---|---|
| | | CLIP | LEAF | CLIP | LEAF |
| 1 | CLIP | $27.53_{(\pm 4.52)}$ | $\mathbf{28.00}_{(\pm 3.70)}$ | $65.38_{(\pm 12.72)}$ | $66.73_{(\pm 11.61)}$ |
| | LEAF | $\mathbf{28.84}_{(\pm 3.49)}$ | $27.96_{(\pm 3.48)}$ | $\mathbf{69.47}_{(\pm 11.05)}$ | $\mathbf{68.01}_{(\pm 11.17)}$ |
| 2 | CLIP | $22.96_{(\pm 5.80)}$ | $24.46_{(\pm 4.86)}$ | $57.21_{(\pm 13.90)}$ | $60.80_{(\pm 12.52)}$ |
| | LEAF | $\mathbf{26.72}_{(\pm 4.23)}$ | $25.23_{(\pm 4.36)}$ | $\mathbf{66.11}_{(\pm 11.88)}$ | $\mathbf{63.40}_{(\pm 11.95)}$ |
| 3 | CLIP | $19.45_{(\pm 5.86)}$ | $21.30_{(\pm 5.44)}$ | $51.55_{(\pm 13.40)}$ | $55.68_{(\pm 12.59)}$ |
| | LEAF | $\mathbf{24.61}_{(\pm 5.19)}$ | $22.59_{(\pm 5.16)}$ | $\mathbf{62.68}_{(\pm 12.57)}$ | $\mathbf{59.02}_{(\pm 12.19)}$ |
| 4 | CLIP | $17.42_{(\pm 5.68)}$ | $19.10_{(\pm 5.48)}$ | $48.34_{(\pm 12.66)}$ | $52.24_{(\pm 12.20)}$ |
| | LEAF | $\mathbf{22.44}_{(\pm 5.78)}$ | $20.25_{(\pm 5.44)}$ | $\mathbf{59.25}_{(\pm 12.95)}$ | $\mathbf{55.36}_{(\pm 12.33)}$ |

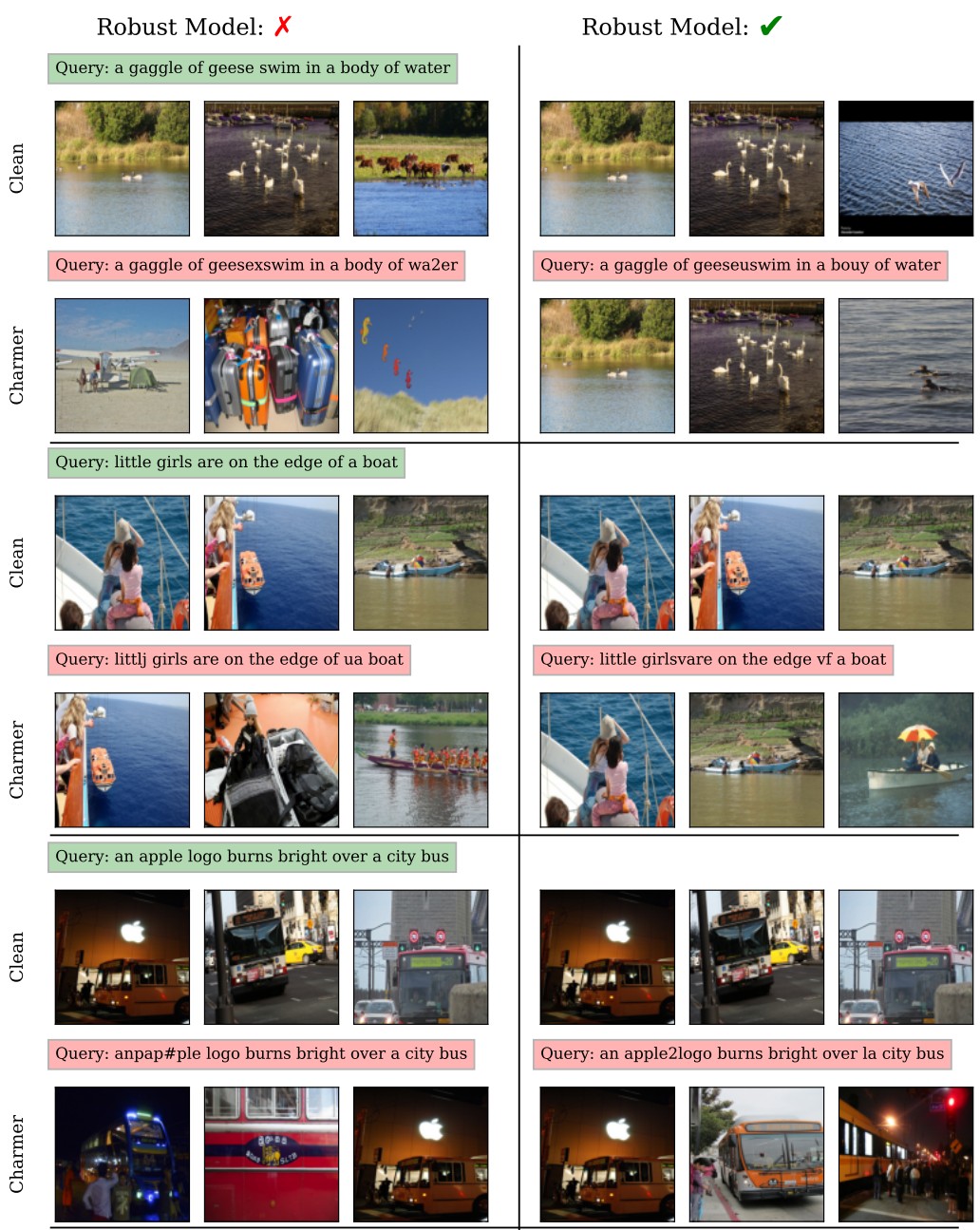

Figure 10: **Visualizing MS-COCO retrieved images.** For our ViT-L/14 robust model and it's non-robust counterpart, we show the top-3 retrieved images for the original Query and the perturbed Query via the constrained Charmer ($k = 2, n = 10$) attack. On average, the robust model is able to preserve the order and retrieves semantically relevant images (esp. top-1) even under perturbation.

Table 21: **Performance of SD-1.5 under typographic attacks:** The generation quality is low with both the original CLIP text encoder and the LEAF counterpart. As a reference, the generation quality of SD-1.5 with unperturbed inputs is a CLIPScore of 31.50 T2I and 73.31 I2I. However, LEAF is able to attain a higher score both in T2I and I2I CLIPScore.

| Text encoder | CLIPScore T2I | CLIPScore I2I |
|---|---|---|
| CLIP | $16.79_{(\pm 4.63)}$ | $45.27_{(\pm 13.12)}$ |
| LEAF | $\mathbf{17.41}_{(\pm 4.27)}$ | $\mathbf{48.04}_{(\pm 13.25)}$ |

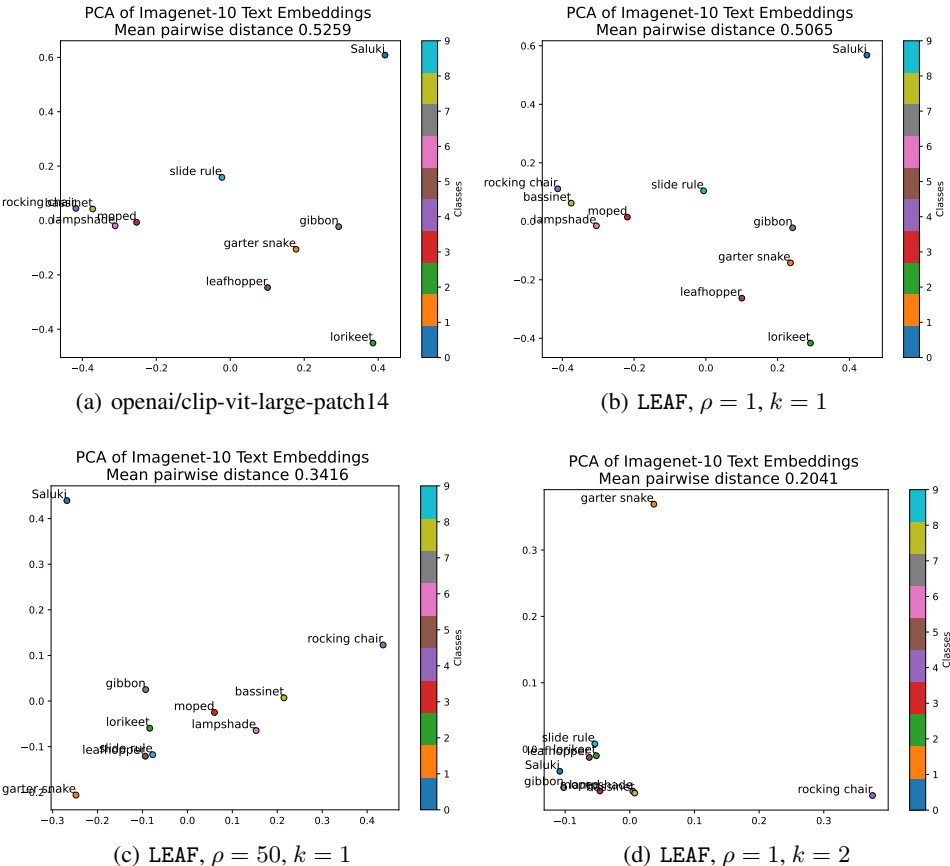

Figure 11: Ablation study on the cause of the clean performance drop in zero-shot classification.

## D.6   Ablation studies

In Appendix D.6.1 we evaluate the performance drop in zero-shot image classification when training without semantic constraints. In Appendix D.6.2 we measure the Eq. (TextFARE) loss before and after training.

### D.6.1   On the performance drop without semantic constraints

First, we perform an ablation study to better understand the cause of the performance drop in terms of clean accuracy in Table 8. We select 10 classes from the ImageNet dataset and visualize the corresponding text embeddings using the prompt "a photo of a LABEL". In Fig. 11, we observe that as $\rho$ and $k$ increase, the class projections in 2D space become more clustered. We compute the mean pairwise distance, defined as the average L2 distance between all class pairs, and find that it decreases significantly.

Table 22: **Text embedding inversion examples for ViT-H/14.** We highlight in `red` words that are reconstructed by the robust model but not by the clean model; in `teal` words that are reconstructed by the clean model but not by the robust model; and in `yellow` words that are not reconstructed by either model. The robust model clearly misses fewer words.

| Original | Robust | Reconstructed ViT-H/14 |
|---|---|---|
| A car `and` a `public` transit vehicle `on` a road. | ✗ | public transit car alongside a vehicle amongst partially road road ." |
| | ✓ | jrnotified car and transit vehicle sit on a road ). |
| `An` `image` `of` `a` `hotel` bath-room `that` `is` ugly. | ✗ | ugly bathroom demonstrating poorly gross envir[U+0442]khobbhutto? |
| | ✓ | ugly hotel bathroom showcasing concerns resemble ?magbbhutto. |
| `An` `older` `picture` `of` `a` large kitchen `with` `white` `appliances.` | ✗ | older earliest appenhistorical archival picture featuring older smaller large kitchen |
| | ✓ | large kitchen pictured prior a a looked white appliances unidenti). |
| `A` girl sitting `on` `a` bench `in` `front` `of` `a` stone wall. | ✗ | prepped amina ssels sitting sitting bench near stone textured wall [U+1F91F]girl girl |
| | ✓ | laghateparth girl twitart bench sitting outside a stone wall ??▷." |
| `A` clean kitchen `with` `the` windows `white` `and` `open.` | ✗ | behold beautiful windows bein somewhere '; white-beautifully clean kitchen |
| | ✓ | a a kitchen with windows white wit yet clean . |
| Two women waiting `at` `a` bench `next` `to` `a` street. | ✗ | ¡/ ¡/ ": ¡/ ,' two women waiting bench against street |
| | ✓ | : two women waiting at an street bench ?bbcone . |
| `An` `office` `cubicle` `with` four `different` `types` `of` `computers.` | ✗ | four various computically cubicè compu?their desktop desk parked |
| | ✓ | office cubic??eczw with four different computers either |
| `An` `old` victorian `style` bed `frame` `in` `a` `bedroom.` | ✗ | old ornate victorian bed showcasing ?wouldfeeold ). |
| | ✓ | victorian finornate bed frame placed in a bedroom . |
| A `striped` plane `flying` `up` `into` the `sky` `as` the `sun` `shines` `behind` `it.` | ✗ | a sized ¡/ wildly crafted plane near dramatically dramatically sun sunlight stripes approaching upward underneath |
| | ✓ | a striped ??ûp plane coming above into sun ?[U+0648]sky . |
| A cat `in` `between` two cars `in` `a` parking `lot.` | ✗ | seemingly domestic cat sits standing among two cars in parking %. |
| | ✓ | cat between two ?four cars docked paved parking lot . |

Table 23: **Text embedding inversion examples for ViT-g/14.** We highlight reconstructions, we highlight in red words that are reconstructed by the robust model but not by the clean model; in teal words that are reconstructed by the clean model but not by the robust model; and in yellow words that are not reconstructed by either model. The robust model clearly misses fewer words.

| Original | Robust | Reconstructed ViT-H/14 |
|---|---|---|
| A car and a public transit vehicle on a road. | ✗ | partially tionally car sits alongside alongside roads public transit vehicle '. |
| | ✓ | a car and eachother and a roadway public transit vehicle . |
| An image of a hotel bathroom that is ugly. | ✗ | apparent nicely tered hotel bathroom containing looking ugly pfmage |
| | ✓ | image of a ugly と繋?* an hotel bathroom . |
| An older picture of a large kitchen with white appliances. | ✗ | a large kitchen photographed before that wasn resembtedly older . |
| | ✓ | large old whil, an kitchen featuring ☀reaswhite appliances |
| A girl sitting on a bench in front of a stone wall. | ✗ | girl near stone wall in a bench aciantly sitting tedly tedly ). |
| | ✓ | girl sitting while a stone wall sits alongside an bench a ¡end_of_text¿). |
| A clean kitchen with the windows white and open. | ✗ | view of a white kitchen and nicely clean windows . |
| | ✓ | an clean and white kitchen with windows thwindows . |
| Two women waiting at a bench next to a street. | ✗ | along a street bench . two women crouwaited stares . |
| | ✓ | :// ; two women wait a street while bench outside . |
| An office cubicle with four different types of computers. | ✗ | office cubicle depicting four various different computers alongside paysoff ). |
| | ✓ | office cubicle containing an workplace with four different types computers |
| An old victorian style bed frame in a bedroom. | ✗ | eighsundaymotivation throwback© ?shutterintimacy "; victorian bed |
| | ✓ | a victorian style bed frame uas in a bedroom . |
| A striped plane flying up into the sky as the sun shines behind it. | ✗ | nearly seemingly seemingly oooooooo a striped ambitious plane being flying into sky with sun light |
| | ✓ | a striped plane being flying over above , but shining sun enguliot ung behind |
| A cat in between two cars in a parking lot. | ✗ | cat sitting through parked parking lot ?) alongside two two cars |
| | ✓ | cat sits in an parking lot between two cars either ). |

Table 24: **Detailed retrieval results for** $k = 2$, $n = 10$ **constrained attack.** This is an extension of Table 3 for the ViT-L/14 model. We show how the robustness changes with changing training $\rho$ across the three target texts.

| | | MS-COCO T→I retrieval | | | |
| | Train | Clean | | Charmer-Con | |
| Model | $\rho$ | R@1 | R@5 | R@1 | R@5 |
|---|---|---|---|---|---|
| Target: `A man aggressively kicks a stray dog on the street.` | | | | | |
| non-robust | - | 49.11 | 73.79 | 28.88 | 52.58 |
| CLIP-ViT-L/14 | 1 | 49.33 | 73.98 | 37.34 | 62.16 |
| CLIP-ViT-L/14 | 2 | 49.35 | 73.73 | 37.78 | 62.84 |
| CLIP-ViT-L/14 | 5 | 49.63 | 73.82 | 38.66 | 63.86 |
| CLIP-ViT-L/14 | 10 | 48.99 | 73.60 | 40.22 | 65.30 |
| CLIP-ViT-L/14 | 20 | 48.97 | 73.72 | 37.92 | 62.44 |
| CLIP-ViT-L/14 | 50 | 48.71 | 73.72 | 40.70 | 66.20 |
| Target: `This is an image of a a pyramid.` | | | | | |
| non-robust | - | 49.11 | 73.79 | 31.90 | 54.90 |
| CLIP-ViT-L/14 | 1 | 49.33 | 73.98 | 36.30 | 60.08 |
| CLIP-ViT-L/14 | 2 | 49.35 | 73.73 | 39.55 | 64.65 |
| CLIP-ViT-L/14 | 5 | 49.63 | 73.82 | 40.38 | 65.34 |
| CLIP-ViT-L/14 | 10 | 48.99 | 73.60 | 37.60 | 62.20 |
| CLIP-ViT-L/14 | 20 | 48.97 | 73.72 | 40.00 | 65.46 |
| CLIP-ViT-L/14 | 50 | 48.71 | 73.72 | 41.42 | 66.66 |
| Target: `A group of teenagers vandalizes a public statue.` | | | | | |
| non-robust | - | 49.11 | 73.79 | 30.68 | 54.22 |
| CLIP-ViT-L/14 | 1 | 49.33 | 73.98 | 35.26 | 59.36 |
| CLIP-ViT-L/14 | 2 | 49.35 | 73.73 | 39.29 | 63.76 |
| CLIP-ViT-L/14 | 5 | 49.63 | 73.82 | 36.74 | 61.36 |
| CLIP-ViT-L/14 | 10 | 48.99 | 73.60 | 41.42 | 65.50 |
| CLIP-ViT-L/14 | 20 | 48.97 | 73.72 | 41.04 | 66.12 |
| CLIP-ViT-L/14 | 50 | 48.71 | 73.72 | 38.56 | 62.38 |

Table 25: **Bimodal attacks in MS-COCO text-to-image retrieval**. Following [Zhang et al., 2022], we attack the vision-only robust (FARE) and our bimodally robust LEAF models. First we attack the text modality with Charmer-Con ($k = 1$) and then use APGD with 100 iterations to perturb input images.

| | Recall@1 | | | Recall@5 | | |
| Method | Clean | $\epsilon = \frac{2}{255}, k = 1$ | $\epsilon = \frac{4}{255}, k = 1$ | Clean | $\epsilon = \frac{2}{255}, k = 1$ | $\epsilon = \frac{4}{255}, k = 1$ |
|---|---|---|---|---|---|---|
| Original | 48.9 | 17.2 | 8.9 | 73.1 | 35.2 | 19.7 |
| FARE | 49.1 | 36.6 | 35.8 | 73.8 | 62.2 | 61.0 |
| LEAF | 48.7 | 43.4 | 42.8 | 73.7 | 67.4 | 66.9 |

#### D.6.2 On the Eq. (`TextFARE`) loss

In this section, we evaluate the effectiveness of our method `LEAF` in minimizing the loss in Eq. (`TextFARE`). First, we measure the loss before and after adversarial finetuning in the ViT-L/14 scale on the first 100 images in the AG-News dataset at $k = 1$. We evaluate the inner max of Eq. (`TextFARE`) with the `LEAF` attack with and without semantic constraints (Appendix D.1) and with $\rho \in \{1, 2, 5, 10, 20, 50\}$. As baselines, we evaluate the same term with the Charmer-20 attack and a Bruteforce approach, which evaluates all of the possible sentences at Levenshtein distance $k = 1$.

In Fig. 12 we can observe that training with `LEAF`, we generalize to be robust to stronger attacks, even if they do not employ semantic constraints. For all cases, employing a larger $\rho$ reduces the

Table 26: **Evaluating the loss in Eq. (FARE) and Eq. (`TextFARE`) across different scales:** We evaluate the ViT-L/14, ViT-H/14 and ViT-g/14 with and without our adversarial finetuning (LEAF) in both the image (ImageNet) and text domain (AG-News). $L_{\text{clean}}$ refers to the respective loss when there is no perturbation applied, thus measuring the deviation to the original model. Robust models present a lower adversarial loss in both domains, with larger models presenting a higher loss before and after adversarial finetuning due to the use of larger embedding dimensions.

| Model | Robust | ImageNet | | AG-News | | |
|---|---|---|---|---|---|---|
| | | $L_{\text{clean}}$ | $L_{\text{adv}}$ | $L_{\text{clean}}$ | $L_{\text{adv-cons.}}$ | $L_{\text{adv-uncons.}}$ |
| ViT-L/14 | ✗ | 0.0 | 789.7 | 0.0 | 58.4 | 82.6 |
| ViT-L/14 | ✓ | 33.1 | 56.4 | 6.8 | 23.6 | 41.7 |
| ViT-H/14 | ✗ | 0.0 | 1042.8 | 0.0 | 73.4 | 111.3 |
| ViT-H/14 | ✓ | 47.9 | 89.6 | 13.3 | 40.7 | 76.3 |
| ViT-g/14 | ✗ | 0.0 | 2172.5 | 0.0 | 112.3 | 175.0 |
| ViT-g/14 | ✓ | 93.6 | 181.2 | 18.8 | 66.0 | 121.6 |

gap between the `LEAF` estimate and the true inner max of Eq. (`TextFARE`), i.e., Bruteforce. After adversarial finetuning with `LEAF`, both the loss estimates with Charmer-20 and Bruteforce are reduced.

Then, we evaluate the inner max of Eq. (`TextFARE`) in the ViT-L/14, ViT-H/14 and ViT-g/14 scales with Charmer-20 before and after adversarial finetuning with `LEAF`. Similarly, the Charmer-20 loss is minimized even if no semantic constraints are used in the estimate, for all model sizes. The loss is larger for larger model sizes both before and adversarial finetuning. This could be due to the larger embedding dimension for the ViT-H/14 and ViT-g/14 models. Finally, we also evaluate the inner max of Eq. (FARE) in the image domain. To this end, we compute adversarial perturbations for 100 ImageNet images with a 100-steps APGD attack on the Eq. (FARE) objective at radius $\epsilon = 2/255$. The results are reported in Table 26: similar to the textual attacks, we observe that the loss increases with model size. Importantly, the robust models generally demonstrate much smaller adversarial loss than their original counterparts. These results validate the intuition from Fig. 1 (left): the robust models map perturbed inputs much closer to the original inputs than the original models.

### D.6.3 Performance under token-level attacks

In this section, we evaluate the performance of our `LEAF` ViT-L/14, ViT-H/14 and ViT-g/14 models under the TextFooler token-level adversarial attack [Jin et al., 2020]. Furthermore, we replicate the experiment by Abad Rocamora et al. [2024] and finetune BERT-base [Devlin et al., 2019] on the SST-2 dataset with our character-level attack to evaluate the character-level and token-level accuracy of the classifier.

Abad Rocamora et al. [2024] conclude that token-level defenses are not effective for character-level attacks and vice-versa. In Table 28, we can observe that the in line with their results, character-level defenses are not effective for the token-level TextFooler attack.

In Table 27 we present the BERT-base Adversarial Training results. In line with the results of [Abad Rocamora et al., 2024], we observe that adversarial training with character-level attacks does not improve the robustness in the token level. Regarding character-level robustness, we observe that `LEAF` obtains almost 5 points less in adversarial accuracy with respect to training with Charmer, but preserves a clean accuracy 4 points higher.

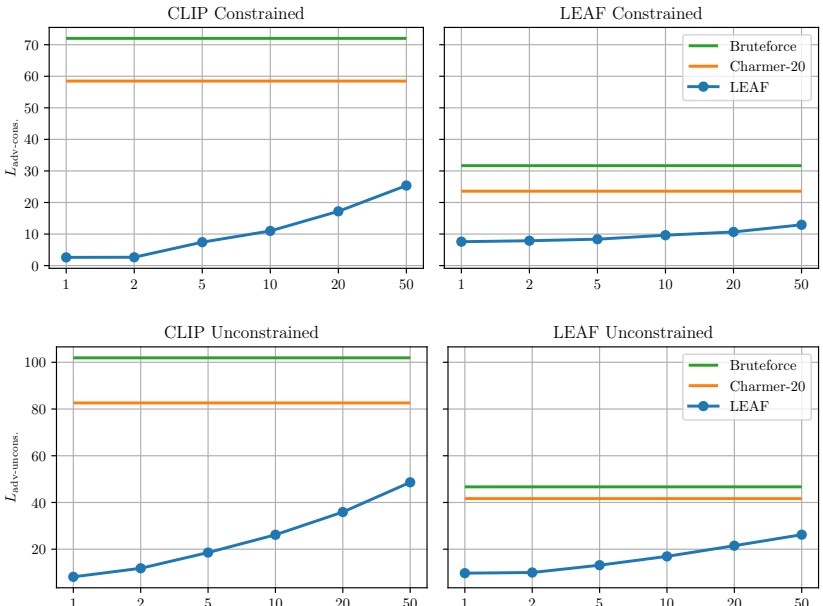

Figure 12: **Evaluating the loss in Eq. (`TextFARE`) with different attacks:** We evaluate the models in the ViT-L/14 scale on the first 100 sentences in the AG-News test dataset. For increasing values of $\rho$, the `LEAF` attack approximates better the inner max in Eq. (`TextFARE`), getting closer to the Bruteforce maximum. Our models, trained with `LEAF` and $\rho = 50$, reduce the Bruteforce loss, meaning that our models generalize to stronger attacks.

Table 27: **Adversarial Training of BERT-base models in SST-2:** We report the clean accuracy, character-level (Charmer) adversarial accuracy and token-level (TextFooler) adversarial accuracy.

| Method | Acc. | Adv. (Charmer) | Adv. (TextFooler) |
|---|---|---|---|
| Original* | **92.43** | 33.26 | 4.47 |
| TextGrad* [Hou et al., 2023] | 80.94 | 26.44 | **23.18** |
| Charmer* [Abad Rocamora et al., 2024] | 87.20 | **69.46** | 4.21 |
| LEAF | 91.51 | 64.68 | 5.50 |
| LEAF-constrained | 91.86 | 62.27 | 4.13 |

∗ Numbers from Abad Rocamora et al. [2024]. The results were obtained as an average of 5 training runs.

Table 28: **Token-level adversarial attacks in zero-shot text classification.** We report the TextFooler adversarial accuracy (Adv.) on on AG-News and SST-2.

| Model | Robust | AG-News Acc. | AG-News Adv. | SST-2 Acc. | SST-2 Adv. |
|---|---|---|---|---|---|
| CLIP-ViT-L/14 | ✗ | 74.4 | 1.70 | 71.2 | 0.57 |
| | ✓ | 78.0 | 1.70 | 71.9 | 0.80 |
| OpenCLIP-ViT-H/14 | ✗ | 71.1 | 1.60 | 61.6 | 1.83 |
| | ✓ | 72.3 | 1.00 | 58.4 | 2.98 |
| OpenCLIP-ViT-g/14 | ✗ | 67.3 | 0.50 | 57.8 | 1.83 |
| | ✓ | 66.7 | 1.20 | 56.0 | 3.10 |

