# OpenReview forum: "Robustness in Both Domains: CLIP Needs a Robust Text Encoder"
_NeurIPS.cc/2025/Conference — NeurIPS 2025 poster_

### Official Review · Reviewer_qiLf · 2025-06-27

**Clarity:** 4
**Significance:** 3
**Originality:** 3
**Rating:** 4
**Confidence:** 4

**Summary:**

The paper addresses the adversarial robustness of CLIP's text encoder, a relatively unexplored area compared to the image encoder. The authors propose LEAF, an efficient adversarial finetuning method. The core idea is to finetune the text encoder to produce embeddings for adversarially perturbed text (via character-level changes) that remain close in L2 distance to the embeddings of the original, clean text. The contributions are demonstrated across several downstream tasks, showing that this approach improves robustness in zero-shot classification, text-to-image retrieval, and the quality of text-to-image generation models under textual adversarial attack.

**Questions:**

The primary concern is the scope of the evaluation. Could the authors comment on the feasibility of extending their evaluation to a wider suite of zero-shot classification benchmarks commonly used for CLIP (e.g. VTAB, https://github.com/LAION-AI/CLIP_benchmark, etc.)? Showing consistent gains across a more diverse set of data distributions would significantly strengthen the paper's claims.

The paper focuses on character-level attacks. Have the authors considered how LEAF might perform against token-level attacks, which aim to preserve semantics while changing the embedding? Even a preliminary experiment or a discussion on why LEAF might or might not be suited for this threat model would be insightful.

Regarding the semantic constraints used during training: The results in Figure 3 suggest these are critical to prevent performance degradation on the image task. Could you elaborate on their necessity and how sensitive the model's performance is to the strictness of these rules? Does this imply that for new domains or languages, new hand-crafted constraints would be required?

**Ethical Concerns:**

["NO or VERY MINOR ethics concerns only"]

**Final Justification:**

The paper makes a clear and well-supported contribution to the underexplored area of CLIP text encoder robustness. The rebuttal satisfactorily addressed my main concerns with additional experiments and clarifications. While some scope limitations remain, the strengths outweigh them, so I am maintaining my score.

**Limitations:**

Yes, the authors commendably discuss limitations such as the decoupled training paradigm (and the lack of defense against joint attacks) and the focus on character-level attacks. The discussion is fair and accurate. To further improve, it could be beneficial to also explicitly frame the limited set of evaluation benchmarks and model scales as a scope limitation of the current study in the limitations section.

**Paper Formatting Concerns:**

No formatting concerns.

**Quality:**

3

**Strengths And Weaknesses:**

Strengths:
- Significance and Originality: The paper tackles a clear, important, and largely overlooked problem. Ensuring the robustness of the text encoder is crucial for the reliability of the many downstream models that leverage CLIP. This work is one of the first to provide a systematic study and solution in this domain.
- Methodological Quality: The proposed method, LEAF, is well-motivated and pragmatic. It is designed for computational efficiency, which is a significant advantage in the context of adversarial training. The decoupled training approach, focusing only on the text encoder with a simple L2 objective, makes the method practical and scalable.
- Experimental Breadth and Clarity: The authors demonstrate the value of their approach across three distinct and important use cases of CLIP (classification, retrieval, generation). This multi-faceted evaluation makes the practical benefits of the work very clear. The paper is exceptionally well-written, with figures (e.g., Fig. 1 and Fig. 5) that effectively communicate the core ideas and results.

Weaknesses:
- Limited Evaluation Scope: This is the primary weakness. While the experiments are of good quality, their scope is narrow. For zero-shot classification, robustness is only shown on a couple of datasets. A standard CLIP evaluation would involve a much wider suite of benchmarks (e.g. VTAB etc.), which would be necessary to make a stronger claim about general robustness. The evaluation on different model scales could also be more extensive.

- Constrained Threat Model: The work focuses exclusively on character-level attacks. While this is an important threat, it overlooks other potential vulnerabilities, such as token-level attacks (e.g., replacing words with synonyms). The robustness of the proposed method against these other attack types is unknown.

- Reliance on Heuristics: The method's success without degrading vision performance relies on "semantic constraints" during training (i.e., ensuring perturbations do not form new valid words). This indicates that some degree of task-specific, hand-crafted rules are needed, making the solution slightly less general than a fully automated approach.

---

> ### Author Rebuttal · Authors · 2025-07-29
>
> Dear Reviewer qiLf,
>
> Thanks for your time reviewing our paper. We are happy you found our paper exceptionally well written. We answer to your comments as follows:
>
> ## P1: “A standard CLIP evaluation would involve a much wider suite of benchmarks (e.g. VTAB etc.)”
>
> We agree that CLIP models should be evaluated on a wide suite of tasks. Thank you for suggesting the VTAB benchmark. We report in Table 9 (appendix) the performance of our models and baselines across several zero-shot classification tasks, many of which are part of the VTAB benchmark. Additionally, we have evaluated the models on the remaining VTAB subtasks and report averaged performance over the categories “natural”, “specialized”, and “structured” in Table A below. We observe that in clean evaluation, robust models sacrifice performance on “natural” and “specialized” (a trade-off between clean and robust performance is expected [1]). On “structured” the behavior is mixed - sometimes even outperforming the non-robust models. In the adversarial evaluation ($\epsilon=\frac{2}{255}$), we observe that the non-robust models are completely vulnerable, while our robust models maintain much better performance when attacked. We will add these interesting results to the appendix of the manuscript.
>
> Table A: Clean and adversarial evaluation on VTAB.
>
> | attack    | model             | robust | natural | specialized | structured |
> |:----------|:------------------|:-------|--------:|------------:|-----------:|
> | clean     | CLIP-ViT-L/14     | ❌      |    74.4 |        63.5 |       11.9 |
> |           |      | ✅      |    68.5 |        41.9 |       13.3 |
> |           | OpenCLIP-ViT-H/14 | ❌      |    78.7 |          57.0 |       11.7 |
> |           |  | ✅      |    74.8 |        45.6 |       11.8 |
> |           | OpenCLIP-ViT-g/14 | ❌      |    79.5 |        62.9 |       12.5 |
> |           |  | ✅      |    72.4 |        51.4 |       11.4 |
> | $\epsilon=\frac{2}{255}$ | CLIP-ViT-L/14     | ❌      |       0.0 |           0.0 |          0.0 |
> |           |  | ✅      |    42.4 |        10.6 |        3.9 |
> |           | OpenCLIP-ViT-H/14 | ❌      |     0.1 |           0.0 |          0.0 |
> |           |  | ✅      |    44.9 |        14.6 |        3.6 |
> |           | OpenCLIP-ViT-g/14 | ❌      |       0.0 |           0.0 |          0.0 |
> |           |  | ✅      |      41.0 |         9.5 |        1.9 |
>
>
>
> ## P2: “The evaluation on different model scales could also be more extensive”
>
> Please note that Tables 2,3,4 and 9 include evaluations in all ViT-L, H and g sizes. We even finetuned the text encoder of the ViT-bigG 3B parameter model and evaluated it on text-to-image generation. We agree that it would be nice to evaluate models larger than this, but our compute budget does not allow for larger sizes, this is indeed a limitation as we mention in lines 283-284.
>
> ## P3: “Have the authors considered how LEAF might perform against token-level attacks, which aim to preserve semantics while changing the embedding?”
>
> While token-level attacks can also be realistic, recent studies show that token-level attacks often change the semantics of the sentence, leading to invalid attacks [2,3,4]. Moreover, **[5] show that token-level defenses are not effective for character-level adversarial attacks and vice-versa.** For completeness, we have evaluated the performance of our CLIP models in zero-shot text classification, under the token-level TextFooler attack [6].
>
> Table B: Adversarial accuracy in AG-News and SST-2 under the TextFooler attack.
> |            |               | AG-News         | SST-2           |
> |------------|---------------|-----------------|-----------------|
> | Model size | Model         | TextFooler Adv. | TextFooler Adv. |
> | L          | CLIP | 1.70            | 0.57            |
> |            | LEAF          | 1.70            | 0.80            |
> | H          | OpenCLIP | 1.60            | 1.83            |
> |            | LEAF          | 1.00            | 2.98            |
> | g          | OpenCLIP | 0.50            | 1.83            |
> |            | LEAF          | 1.20            | 3.10            |
>
> Our results are in line with the findings of [5]: Character-level defenses are not effective for token-level adversarial attacks.
>
> ## P4: Could you elaborate on the need for semantic constraints during training? If a new language/task is employed, should the constraints change?
>
> Thanks for bringing up this very interesting point. We believe that the semantic constraints we employ are general enough and should not change from task to task. In fact, the same constraints used during training provided a good performance in a wide range of tasks (text classification, text-to-image retrieval, text-to-image generation and text embedding inversion). Moreover, in Fig. 11, we report the loss with and without constraints. Training with constraints, also generalizes to reduce the unconstrained loss.
>
> The case of changing languages is an interesting area for future work. Since the constraints we employ during training are based on the English dictionary, the performance on datasets in other languages might not be so good. However, since the model is able to generalize and be robust to unconstrained attacks, we believe that robustness will (to some extent) generalize from language to language.
>
> Thanks again for your suggestions. We will incorporate the new experiments and discussions in the revised version of the manuscript. If you have further comments, we will be happy to answer. If our comments addressed your concerns, we would appreciate a raise in the rating.
>
> Best regards,
>
> Authors.
>
>
> **References**
>
> [1] Dimitris Tsipras et al., Robustness May Be at Odds with Accuracy, ICLR 2019
>
> [2] John Morris et al., Reevaluating adversarial examples in natural language. In Findings of the Association for Computational Linguistics: EMNLP 2020.
>
> [3] Salijona Dyrmishi et al., How do humans perceive adversarial text? a reality check on the validity and naturalness of word-based adversarial attacks. In Proceedings of the 61st Annual Meeting of the Association for Computational Linguistics (Volume 1: Long
> Papers), 2023.
>
> [4] Bairu Hou et al., Textgrad: Advancing robustness evaluation in NLP by gradient-driven optimization. In International Conference on Learning Representations (ICLR), 2023.
>
> [5] Elias Abad Rocamora et al., Revisiting Character-level Adversarial Attacks for Language Models, International Conference on Machine Learning, 2024.
>
> [6] Di Jin et al., Is bert really robust? a strong baseline for natural language attack on text classification and entailment. AAAI Conference on Artificial Intelligence, 2020.

---

### Official Review · Reviewer_qWrN · 2025-06-30

**Clarity:** 3
**Significance:** 2
**Originality:** 2
**Rating:** 3
**Confidence:** 3

**Summary:**

This paper extends the FARE objective—originally developed for adversarial robustness in CLIP image encoders—to the text domain, introducing LEAF: an efficient character-level adversarial finetuning method for CLIP text encoders. By sampling a limited number of perturbations per batch, LEAF achieves comparable robustness to Charmer while being significantly faster. Extensive evaluation spans zero-shot text classification (AG-News), multimodal retrieval, text-to-image generation (using SD-1.5 and SDXL), and embedding inversion, showing substantial robustness gains with minimal clean performance degradation.

**Questions:**

- Given the growing importance of text precision in T2I models, have you evaluated LEAF’s effect on intended typographic prompts, such as rare text content (“a ST00P sign”)? Does enforcing adversarial robustness inadvertently harm those cases?
- In Table 14-16, were the same adversarial captions used for baseline and LEAF models? If they were generated independently, can you clarify how fairness in comparison is maintained?

**Ethical Concerns:**

["NO or VERY MINOR ethics concerns only"]

**Limitations:**

yes

**Paper Formatting Concerns:**

No concerns

**Quality:**

3

**Strengths And Weaknesses:**

Strengths
-  Implements an elegant, easy-to-reuse adversarial objective with a fast sampling attack (LEAF), achieving strong robustness with ~5–10× speedup over Charmer.
- Applies analyses across diverse tasks—classification, retrieval, generation, inversion—demonstrating robustness gains throughout.
- Validates LEAF on large backbones (ViT-H/14, ViT-g/14) effectively.

Weaknesses
- The core contribution is adapting FARE to text, with the primary novelty being a faster attack; the methodological lift remains incremental. The approach handles only character-level perturbations; broader attacks (token-level, syntactic, semantic) remain unexplored.
- Text and image encoders are finetuned separately, leaving effectiveness under joint attacks untested.
- For text-to-image evaluations, only SD-1.5 and SDXL are tested. Stronger, state-of-the-art generation architectures such as Stable Diffusion 3 (SD3) or Flux.1 are not evaluated, limiting insight into broader applicability
- Appendix Table 14-16 suggests that adversarial captions differ across models when comparing baselines. If baseline and robust models are evaluated on different perturbed prompts, the comparison may be unfair.

---

> ### Author Rebuttal · Authors · 2025-07-29
>
> Dear Reviewer qWrN,
>
> We appreciate the time dedicated to reviewing our paper. We are happy that you describe our method (LEAF) as elegant and easy-to-reuse, and that you find our evaluation diverse and effective. We address your concerns in the following:
>
> ## P1: “The core contribution is adapting FARE to text, with the primary novelty being a faster attack; the methodological lift remains incremental.”
>
> The methodological lift includes a new efficient training-time attack and a principled constrained objective for the discrete text modality. Moreover, our study is the first demonstration of dual-domain robustness in CLIP. Our experiments reveal enhanced robustness to character-level attacks in several tasks and improved interpretability of robust models via embedding inversion.
>
> ## P2: “The approach handles only character-level perturbations; broader attacks (token-level, syntactic, semantic) remain unexplored.”
>
> Robustness in the text domain can be studied through many threat models. As mentioned in lines 94-97, token-level attacks often alter the semantics of the sentence [1,2,3]. Moreover, **[8] show that token-level defenses are not effective against character-level attacks and vice-versa.** To keep our insights clear, we decided to focus on character-level attacks. Nevertheless, for completeness, we have evaluated the zero-shot text adversarial accuracy of our CLIP models, with the Textfooler token-level adversarial attack [7] in the AG-News and SST-2 datasets.
>
> Table A: Adversarial accuracy in AG-News and SST-2 under the TextFooler attack.
> |            |               | AG-News         | SST-2           |
> |------------|---------------|-----------------|-----------------|
> | Model size | Model         | TextFooler Adv. | TextFooler Adv. |
> | L          | CLIP | 1.70            | 0.57            |
> |            | LEAF          | 1.70            | 0.80            |
> | H          | OpenCLIP | 1.60            | 1.83            |
> |            | LEAF          | 1.00            | 2.98            |
> | g          | OpenCLIP | 0.50            | 1.83            |
> |            | LEAF          | 1.20            | 3.10            |
>
> Our results are in line with the findings of [8]: Character-level defenses are not effective for token-level adversarial attacks. We hope that our work can foster additional studies that enhance the robustness of CLIP under different threat models, e.g. token-level or syntactic attacks.
>
> ## P3: “Text and image encoders are finetuned separately, leaving effectiveness under joint attacks untested.”
>
> As a proof of concept, we have tested the performance of our models when both the image and text are perturbed in text-to-image retrieval.
>
> Starting with the k=1 text perturbations for the ViT-L/14 models from Table 3 in the main paper, we perturb the images with **APGD for 100 iterations with $\epsilon=2/255$ and $\epsilon = 4/255$**. We show text-to-image retrieval on 1k samples from MS-COCO. Specifically, we maximize the distance of the image embeddings under perturbations to the original image. The attack follows a similar setup as CoAttack [6], where the text attack follows the image attack. The clean column here represents the performance of the model when no modality is attacked. In this setup again, our LEAF trained models are the most robust, highlighting the importance of dual modality robustness.
>
> Table B: BiModal attack robustness evaluation for MS-COCO retrieval at $k=1$ and $\epsilon\in \{2, \}4/255$.
> |       | Clean | $\epsilon=\frac{2}{255}$ | $\epsilon=\frac{4}{255}$ | Clean | $\epsilon=\frac{2}{255}$ | $\epsilon=\frac{4}{255}$ |
> |------------------|-------|----------------|-------------------|-------|----------------|-------------------|
> | Method  | Recall@1 | | | Recall@5 | | |
> | Original         | 48.9 | 17.2 | 8.9  | 73.1 | 35.2 | 19.7 |
> | FARE           |   49.1| 36.6 | 35.8 | 73.8| 62.2 | 61.0 |
> | LEAF            | 48.7 | 43.4 | 42.8 | 73.7 | 67.4 | 66.9 |
>
> ## P4: “Stronger, state-of-the-art generation architectures such as Stable Diffusion 3 (SD3) or Flux.1 are not evaluated”
>
> Models like SD3 or FLUX.1 cannot be fully benefited from our approach, since in addition to CLIP text encoders, they employ the encoder-decoder model T5-XXL for obtaining text embeddings. Nevertheless, it is still possible to attack their CLIP text encoders and replace them for our fine-tuned ones, while keeping T5-XXL untouched. We have replicated our text-to-image generation experiments with FLUX.1-dev. Due to the high quality of the generated images (1024x1024), we limit our evaluation to the first 100 images in the MS-COCO dataset.
>
> Table C: Text-to-image and image-to-image CLIP scores for the FLUX.1-dev model with the original (CLIP) and finetuned (LEAF) text encoder.
> | k | Text encoder | CLIPScore T2I  | CLIPScore I2I |
> |---|--------------|----------------|---------------|
> | 0 | CLIP | **30.56**          | **71.19**         |
> |  | LEAF | 30.55          | 71.18         |
> | 1 | CLIP | **29.14**          | 68.09         |
> |  | LEAF | 28.90          | **68.79**         |
> | 2 | CLIP | 27.03          | 63.60         |
> |  | LEAF | **27.38**          | **65.66**         |
> | 3 | CLIP | 24.47          | 59.40         |
> |  | LEAF | **25.71**          | **62.11**         |
> | 4 | CLIP | 22.72          | 57.68         |
> |  | LEAF | **23.51**          | **59.59**         |
>
> Our results show that even when only attacking one text encoder, FLUX.1-dev is still vulnerable to adversarial attacks. When replacing the CLIP-ViT-L text encoder with our LEAF counterpart, the image generation quality is improved for higher values of $k$. Unfortunately, we cannot include image links with our rebuttal. We will include attack examples and generations in the revised version of the manuscript.
>
> We hope our work can motivate the study of the robustness of other architectures like encoder-decoder models, such as T5-XXL.
>
> ## P5: “In Table 14-16, were the same adversarial captions used for baseline and LEAF models? If they were generated independently, can you clarify how fairness in comparison is maintained?”
>
> Fairness relies on the adversarial captions being optimized independently for every model from the same original caption. This procedure yields the adversarial perturbation that maximally degrades the performance of each model. This is a standard practice both in the image and text adversarial robustness communities [4,5]. Attacks optimized over one model and applied to others can also be tested, these are known as transfer attacks. We have tested the adversarial captions optimized over CLIP on LEAF and vice versa for the SD1.5 model.
>
> Table D: Transfer attacks for SD-1.5. We measure the text-to-image CLIPScore. Rows denote target models and columns denote source models.
> | k | Target/Source | CLIP  | LEAF  |
> |---|---------------|-------|-------|
> | 1 | CLIP| 27.53 | 28.00 |
> |   | LEAF| 28.84 | 27.96 |
> | 2 | CLIP| 22.96 | 24.46 |
> |   | LEAF| 26.72 | 25.23 |
> | 3 | CLIP| 19.45 | 21.30 |
> |   | LEAF| 24.61 | 22.59 |
> | 4 | CLIP| 17.42 | 19.10 |
> |   | LEAF| 22.44 | 20.25 |
>
> We observe that, as expected, when the source is equal to the target, the generated image quality is degraded the most. Our text encoder improves the generation quality in all cases except when the Source is LEAF and $k=1$, where CLIP obtains 0.04 points more than LEAF in this advantageous setup.
>
> ## P6: “have you evaluated LEAF’s effect on intended typographic prompts, such as rare text content (“a ST00P sign”)? Does enforcing adversarial robustness inadvertently harm those cases?”
>
> Thanks for the suggestion. As a preliminary test, we have crafted 100 perturbations from the first 100 COCO captions. To take it to the extreme, we have replaced every “i” for a “1”, every “e” for a “3”, every “o” for a “0” and every “a” for an “@”.
>
> For example, the first COCO caption turns into “A w0m@n st@nds 1n th3 d1n1ng @r3@ @t th3 t@bl3.”
>
> Table E: SD1.5 text-to-image generation performance under visually-similar character replacements.
> | Text encoder | CLIPScore T2I | CLIPScore I2I |
> |--------------|---------------|---------------|
> | CLIP| 16.79| 45.27|
> | LEAF| **17.41**| **48.04** |
>
> We observe that while the image generation quality with both encoders is quite low, using LEAF provides an improvement of 0.62 points in CLIPScore T2I and 2.77 in CLIPScore I2I.
>
> Thanks again for your review and suggestions. The added experiments will be included in the revised version of our manuscript. We remain available in case you have additional comments. If you are satisfied with our answers, we would appreciate an increase in the rating.
>
> Best regards,
>
> Authors
>
> **References**
>
> [1] John Morris et al., Reevaluating adversarial examples in natural language. In Findings of the Association for Computational Linguistics: EMNLP 2020.
>
> [2] Salijona Dyrmishi et al., How do humans perceive adversarial text? a reality check on the validity and naturalness of word-based adversarial attacks. In Proceedings of the 61st Annual Meeting of the Association for Computational Linguistics (Volume 1: Long
> Papers), 2023.
>
> [3] Bairu Hou et al., Textgrad: Advancing robustness evaluation in NLP by gradient-driven optimization. In International Conference on Learning Representations (ICLR), 2023.
>
> [4] Francesco Croce, RobustBench: a standardized adversarial robustness benchmark, arXiv preprint arXiv:2010.09670, 2020.
>
> [5] John Morris et al., TextAttack: A Framework for Adversarial Attacks, Data Augmentation, and Adversarial Training in NLP, EMNLP 2020.
>
> [6] Jiaming Zhang et al., Towards adversarial attack on vision-language pre-training models., ACM MM2022
>
> [7] Di Jin et al., Is bert really robust? a strong baseline for natural language attack on text classification and entailment. AAAI Conference on Artificial Intelligence, 2020.
>
> [8] Elias Abad Rocamora et al., Revisiting Character-level Adversarial Attacks for Language Models, International Conference on Machine Learning, 2024.

---

> > ### Comment · Reviewer_qWrN · 2025-08-08
> >
> > Thank you for the detailed rebuttal and additional experiments.
> >
> > - Novelty – While the efficiency improvement is appreciated, I still feel the core contribution—adapting FARE to the text domain—remains incremental, with most of the robustness training paradigm carried over from prior work.
> >
> > - Regarding P6 (typographic prompts) – The character-substitution test is useful, but CLIPScore may not be the best metric to evaluate whether intended typographic content is preserved. OCR-based edit distance or human inspection would better capture whether prompts like “ST00P” are faithfully rendered. Could you clarify if LEAF tends to “correct” such intended variations, which could harm use cases needing exact stylization?

---

> > > ### Author Response · Authors · 2025-08-08
> > >
> > > Dear reviewer qWrN,
> > >
> > > Thanks for your reply.
> > >
> > > Let us remark on the novelty with respect to FARE. It’s true that our method is motivated by FARE and the loss function is shared. Nevertheless, we address text-specific challenges as the efficiency of the adversarial attack and the need to use semantic constraints. We believe these two differences are not obvious. A straightforward adaptation of FARE to the text domain would be to use an existing character-level adversarial attack. This is tested in Table 1, where we train with Charmer and observe a prohibitive training time, motivating a solution. Similarly, the performance degradation without semantic constraints observed in Fig. 3, shows naive adaptations do not work.
> > >
> > > Regarding the “typographic prompts”, we are unsure about the experimental setup you propose, is this an standard experiment in the literature? Could you please refer to a paper we can use to replicate it? In our case, we modified the captions of the COCO dataset, so the expected output is an image with some objects. In this setup, there is no target text that should appear in the image, so OCR metrics cannot be used. However, we agree it would be interesting to evaluate if intended typos are corrected in images where text is generated.
> > >
> > > Best regards,
> > >
> > > Authors

---

### Official Review · Reviewer_fLS5 · 2025-07-01

**Clarity:** 3
**Significance:** 3
**Originality:** 2
**Rating:** 4
**Confidence:** 2

**Summary:**

This paper investigates the overlooked vulnerability of CLIP's text encoder to adversarial perturbations and introduces LEAF (Levenshtein Efficient Adversarial Finetuning), a novel method to improve its robustness. While prior efforts have focused on strengthening the image encoder, this work shows that even small character-level changes in text can drastically shift CLIP’s embeddings and degrade downstream performance. LEAF enhances robustness by training the text encoder to produce stable embeddings under controlled character-level perturbations, guided by semantic constraints to preserve sentence meaning. The method is highly efficient, significantly faster than previous adversarial training approaches, and scales well to large models. Experiments demonstrate that LEAF improves adversarial accuracy in zero-shot text classification, enhances robustness in text-to-image retrieval and generation, and enables more faithful reconstruction of text from embeddings. Overall, the paper highlights the importance of robust text encoders for multi-modal models and provides a practical solution to strengthen CLIP’s resilience in both the image and text domains.

**Questions:**

See the above weakness.

**Ethical Concerns:**

["NO or VERY MINOR ethics concerns only"]

**Final Justification:**

Thanks for your reply. Considering your reply and other reviewers' comments, I tend to maintain my current positive score. Good luck!

**Limitations:**

yes

**Paper Formatting Concerns:**

There are no major formatting issues in this paper.

**Quality:**

3

**Strengths And Weaknesses:**

## Strehgths
* One key strength of this paper lies in its novel focus on the robustness of the CLIP text encoder, an area largely neglected in previous research. By introducing the LEAF method, the authors provide an efficient and scalable solution that significantly improves adversarial robustness with minimal impact on clean performance. LEAF’s design allows it to integrate seamlessly into existing pipelines without the need to retrain the entire model, making it practical for real-world deployment.
* Furthermore, the use of character-level perturbations with semantic constraints ensures that the adversarial training remains realistic and meaningful. The paper also demonstrates the broad applicability of the proposed method across multiple tasks—such as zero-shot classification, text-to-image generation, and retrieval—highlighting the general utility and effectiveness of the approach.

## Weaknesses
* One notable weakness of the paper is that the adversarial robustness is only explored in the character-level perturbation setting, while token-level attacks—which may better reflect real-world adversarial scenarios—are not addressed due to their potential to alter semantics.
* Additionally, the robust image and text encoders are fine-tuned independently, without considering joint multimodal adversarial attacks, which may limit the model’s effectiveness against coordinated cross-modal perturbations.

---

> ### Author Rebuttal · Authors · 2025-07-29
>
> Dear Reviewer fLS5,
>
> Thanks for your time reviewing our paper. We appreciate your appraisal of the practicality and applicability of our training method in CLIP-like models. We answer your points regarding the weaknesses of our work as follows:
>
> ## P1: “token-level attacks—which may better reflect real-world adversarial scenarios—are not addressed due to their potential to alter semantics”
>
> Thanks for highlighting this point. We agree that token-level attacks are prone to altering semantics, this is the reason why we decided to focus on character-level attacks. Moreover, **[2] show that token-level defenses are not effective for character-level adversarial attacks and vice-versa.**
>
> We are unsure about which of the two threat models better captures real world scenarios. Both character-level and token-level attacks can be deemed realistic, the first simulates typos/small changes in the input and the second simulates replacement by synonyms. To complete the analysis, we have evaluated the performance of our CLIP models in zero-shot text classification using the token-level TextFooler attack [1].
>
> Table A: Adversarial accuracy in AG-News and SST-2 under the TextFooler attack.
> |            |               | AG-News         | SST-2           |
> |------------|---------------|-----------------|-----------------|
> | Model size | Model         | TextFooler Adv. | TextFooler Adv. |
> | L          | CLIP | 1.70            | 0.57            |
> |            | LEAF          | 1.70            | 0.80            |
> | H          | OpenCLIP | 1.60            | 1.83            |
> |            | LEAF          | 1.00            | 2.98            |
> | g          | OpenCLIP | 0.50            | 1.83            |
> |            | LEAF          | 1.20            | 3.10            |
>
> Our results are in line with the findings of [2]: Character-level defenses are not effective for token-level adversarial attacks.
>
> ## P2: Your evaluation does not consider joint multimodal adversarial attacks
>
> As a proof of concept, we have tested the performance of our models when both the image and text are perturbed in text-to-image retrieval.
>
> Starting with the k=1 text perturbations for the ViT-L/14 models from Table 3 in the main paper, we perturb the images with **APGD for 100 iterations with $\epsilon= 2/255$ and $\epsilon=4/255$**. We show text-to-image retrieval on 1k samples from MS-COCO. Specifically, we maximize the embedding distance of the image encoder under perturbation to the original image (can be interpreted as an untargeted adversarial attack). The attack follows a similar setup as CoAttack [3], where the text attack follows the image attack. The clean column here represents the performance of the model when no modality is attacked. In this setup again, our LEAF trained models are the most robust, highlighting the importance of dual modality robustness.
>
>
> Table B: BiModal attack robustness evaluation for MS-COCO retrieval at $k=1$ and $\epsilon\in \{2, \}4/255$.
> |       | Clean | $\epsilon=\frac{2}{255}$ | $\epsilon=\frac{4}{255}$ | Clean | $\epsilon=\frac{2}{255}$ | $\epsilon=\frac{4}{255}$ |
> |------------------|-------|----------------|-------------------|-------|----------------|-------------------|
> | Method  | Recall@1 | | | Recall@5 | | |
> | Original         | 48.9 | 17.2 | 8.9  | 73.1 | 35.2 | 19.7 |
> | FARE           |   49.1| 36.6 | 35.8 | 73.8| 62.2 | 61.0 |
> | LEAF            | 48.7 | 43.4 | 42.8 | 73.7 | 67.4 | 66.9 |
>
> Thanks for your review. We remain available in case you have additional comments, we will be happy to answer.
>
> Best regards,
>
> Authors.
>
> **References**
>
> [1] Di Jin et al., Is bert really robust? a strong baseline for natural language attack on text classification and entailment. AAAI Conference on Artificial Intelligence, 2020.
>
> [2] Elias Abad Rocamora et al., Revisiting Character-level Adversarial Attacks for Language Models, International Conference on Machine Learning, 2024.
>
> [3] Jiaming Zhang et al., Towards adversarial attack on vision-language pre-training models., ACM MM2022

---

> > ### Comment · Reviewer_fLS5 · 2025-08-04
> >
> > Thanks for your reply. Considering your reply and other reviewers' comments, I tend to maintain my current positive score. Good luck!

---

> > > ### Author Response · Authors · 2025-08-06
> > >
> > > Thanks for your reply and positive feedback.
> > >
> > > Best,
> > >
> > > Authors

---

### Official Review · Reviewer_owKp · 2025-07-03

**Clarity:** 3
**Significance:** 2
**Originality:** 3
**Rating:** 4
**Confidence:** 3

**Summary:**

This paper explores the lack of research on adversarial robustness of CLIP models for text encoding, although existing literature mainly focuses on the robustness of CLIP image encoders. The authors propose LEAF, a new adversarial fine-tuning method designed specifically for the text domain. LEAF significantly accelerates the adversarial training process while maintaining robustness by leveraging parallelizable character-level perturbations. It enhances zero-shot adversarial accuracy and robustness on downstream tasks, and achieves interpretability through embedding inversion.

**Questions:**

see weaknesses.

**Ethical Concerns:**

["NO or VERY MINOR ethics concerns only"]

**Final Justification:**

This paper is well-written in terms of topic selection, experiments, and writing. The author also addressed my concerns in the subsequent discussion. I think this is a good manuscript.

**Limitations:**

yes

**Quality:**

3

**Strengths And Weaknesses:**

Strengths:

This paper explores an important and relatively unexplored area: adversarial robustness of text encoders in multimodal models, especially in popular architectures such as CLIP, filling a research gap.

The proposed LEAF adversarial fine-tuning method is clear and supported by comprehensive experiments covering multiple downstream tasks. The experiments clearly demonstrate a significant improvement in the robustness of the CLIP text encoder, showing that LEAF outperforms previous baselines such as Charmer in terms of computational efficiency and performance.

The paper is well written and clearly structured. The logical logic of each section is clear, and readers can easily understand the main ideas, methods, and experimental results.

Weaknesses:

While the paper is generally clear, it is sometimes too cumbersome in methodological details, especially when explaining perturbation methods and constraints. A concise summary or an intuitive explanation of the attack mechanism in simpler terms could further improve readability.

The practical application of character-level adversarial robustness improvements may oversimplify real-world scenarios. Real-world adversarial scenarios may also involve a wider range of semantic manipulations (e.g., token-level or phrase-level perturbations), so the broader significance of our findings beyond specific applications may remain theoretical.

While this paper is innovative in the specific context of text encoder robustness in multimodal models, the conceptual foundations of adversarial training (character-level attacks, optimization methods) have been well explored in the prior literature. Furthermore, given the rapid development of multimodal base models, limiting the evaluation and discussion to CLIP-based architectures may limit the broad applicability of our findings.

---

> ### Author Rebuttal · Authors · 2025-07-29
>
> Dear Reviewer owKp,
>
> Thanks for your time reviewing our paper. We appreciate your recognition of the clarity and comprehensiveness of our writing and experiments. We summarize and address your comments as follows:
>
> ## P1: “A concise summary or an intuitive explanation of the attack mechanism in simpler terms could further improve readability.”
>
> Thank you for bringing up this point. Our attack is described in Algorithms 1 and 2 in the appendix. Additionally, Fig. 2 portrays a practical example using $\rho=6$. Let us provide an intuition of the overall procedure:
>
> 1. Sample $\rho$ positions within the sentence.
> 2. Replace/Insert a whitespace in those positions.
> 3. Evaluate the loss of the model on the $\rho$ sentences and select the one with the highest loss.
> 4. Sample $\rho$ characters from the vocabulary.
> 5. Replace the sampled characters in the position selected in step 3.
> 6. Evaluate the loss of the model on the $\rho$ sentences and select the one with the highest loss.
> 7. Go to step 1 until $k$ perturbations are done.
>
> Note that this is the most basic version of our attack: without constraints. To constrain the attack, we filter out the changes producing new words after steps 2 and 5. We hope this intuition eases the understanding of our method. We will clarify the mechanism of our attack in the revised version of the manuscript.
>
> ## P2: “Real-world adversarial scenarios may also involve a wider range of semantic manipulations (e.g., token-level or phrase-level perturbations”
>
> While token-level attacks can also be realistic, recent studies show that token-level attacks often change the semantics of the sentence, leading to invalid attacks [1,2,3]. We highlight this motivation in lines 94-96. Moreover, **[4] show that token-level defenses are not effective for character-level attacks and vice-versa.** Following these studies, we decided to tackle character-level robustness.
>
> For completeness, we have evaluated the performance of our CLIP models in zero-shot text classification using the token-level TextFooler attack [5].
>
> Table A: Adversarial accuracy in AG-News and SST-2 under the TextFooler attack.
> |            |               | AG-News         | SST-2           |
> |------------|---------------|-----------------|-----------------|
> | Model size | Model         | TextFooler Adv. | TextFooler Adv. |
> | L          | CLIP | 1.70            | 0.57            |
> |            | LEAF          | 1.70            | 0.80            |
> | H          | OpenCLIP | 1.60            | 1.83            |
> |            | LEAF          | 1.00            | 2.98            |
> | g          | OpenCLIP | 0.50            | 1.83            |
> |            | LEAF          | 1.20            | 3.10            |
>
> Our results are in line with the findings of [4]: Character-level defenses are not effective for token-level adversarial attacks. We hope that our work can foster additional studies that enhance the robustness of CLIP under different threat models, e.g. token-level or syntactic attacks.
>
> ## P3: The experimental evaluation is limited to CLIP-based models
>
> Thanks for highlighting this point. We believe our evaluation in CLIP-based models is well-representative of the multimodality field. While our training method shows promise for other applications / models where robustness in the text domain is needed, CLIP-like architectures are widely employed in many applications as described in lines 16-21. To complete our analysis with additional architectures, we fine-tune BERT-base on SST-2 for 1 epoch with LEAF. We measure the Charmer adversarial accuracy at $k=1$ and the token-level adversarial accuracy with TextFooler. This is equivalent to the experimental setup in Table 4 in [4].
>
> Table B: Adversarial finetuning of BERT-base on SST-2.
> | Method           | Acc.  | Adv. (Charmer) | Adv. (TextFooler) |
> |------------------|-------|----------------|-------------------|
> | Original         | **92.43** | 33.26          | 4.47              |
> | TextGrad         | 80.94 | 26.44          | **23.18**             |
> | Charmer          | 87.20 | **69.46**          | 4.21              |
> | LEAF             | 91.51 | 64.68          | 5.50              |
> | LEAF-constrained | 91.86 | 62.27          | 4.13              |
>
> In line with the results of [4], we observe that adversarial training with character-level attacks does not improve the robustness in the token level. Regarding character-level robustness, we observe that LEAF obtains almost 5 points less in adversarial accuracy with respect to training with Charmer, but preserves a clean accuracy 4 points higher.
>
> We appreciate your comments, which help us improve the quality of our work. If you have additional comments, we will be happy to answer. If our answers addressed your points, we would appreciate an increase in the rating.
>
> Best regards,
>
> Authors
>
> **References**
>
> [1] John Morris et al., Reevaluating adversarial examples in natural language. In Findings of the Association for Computational Linguistics: EMNLP 2020.
>
> [2] Salijona Dyrmishi et al., How do humans perceive adversarial text? a reality check on the validity and naturalness of word-based adversarial attacks. In Proceedings of the 61st Annual Meeting of the Association for Computational Linguistics (Volume 1: Long
> Papers), 2023.
>
> [3] Bairu Hou et al., Textgrad: Advancing robustness evaluation in NLP by gradient-driven optimization. In International Conference on Learning Representations (ICLR), 2023.
>
> [4] Elias Abad Rocamora et al., Revisiting Character-level Adversarial Attacks for Language Models, International Conference on Machine Learning, 2024.
>
> [5] Di Jin et al., Is bert really robust? a strong baseline for natural language attack on text classification and entailment. AAAI Conference on Artificial Intelligence, 2020.

---

> > ### Comment · Reviewer_owKp · 2025-08-07
> >
> > I appreciate the authors' further explanation. The authors provided additional experiments on different scenarios and models to prove their point. Thus, my main concerns have been mostly addressed. I am happy to maintain my positive rating. To further strengthen the manuscript, I hope the authors include necessary details during this discussion in their final version.

---

### Decision · Program_Chairs · 2025-09-17

**Decision:**

Accept (poster)

**Comment:**

This paper introduces LEAF, an efficient adversarial fine-tuning method that uses parallelizable character-level perturbations to enhance the robustness of CLIP's text encoder. The authors demonstrate that LEAF improves zero-shot adversarial accuracy and maintains robustness on downstream tasks.

The primary strength is addressing the important and under-explored area of adversarial robustness for text encoders in vision-language models. The proposed LEAF method is clear, well-motivated, and backed by comprehensive experiments showing significant gains in both robustness and computational efficiency over baselines.

Initial concerns from reviewers were largely resolved during the rebuttal. One remaining point involved a request for a non-standard experiment, which could not be completed as the reviewer did not provide sufficient details despite the authors' request for clarification.

The strengths of the work outweigh the minor, unresolved point from an unresponsive reviewer.